# Harnessing microbial-derived metabolites in the urinary tract to prevent infection induced catheter encrustation

L. Beryl Guterman [1], Madalyn Motsay[1], Benjamin C. Hunt [1], Aimee L. Brauer[1], Brian S. Learman [1], Mindula K. Wijayahena [2], Alexander C. Hoepker [2,3], Diana S. Aga[2,3], Brittany Francis[1], Beatriz M. Fontoura[4], George L. Donati [4,6], Peter J. Bush[5], Namrata Deka[1] & Chelsie E. Armbruster [1] ✉

*Proteus mirabilis* is a predominant cause of catheter associated urinary tract infection (CAUTI), and a key virulence factor is its urease enzyme which can increase urine pH and form urinary stones, causing catheter blockage and facilitating bacteremia. The only FDA approved urease inhibitor, acetohydroxamic acid (AHA), has side effects that limit its clinical use, necessitating new approaches to target urease activity. We previously discovered that common urinary tract colonizers modulate *P. mirabilis* urease activity via secreted small molecules. In this study, we conduct a metabolomics analysis of six modulatory bacterial species to reveal urease-dampening metabolites. Of 31 candidate metabolites, seven reproducibly decrease *P. mirabilis* urease activity. All seven metabolites dampen urease activity in other urease-positive bacterial species, suggesting conserved targets. Six of the metabolites act via mixed inhibition of the urease enzyme. One metabolite, D-imidazole lactate, exhibits a non-competitive mechanism of urease inhibition along with antimicrobial activity and repression of the urease operon in *P. mirabilis*. Metabolite combinations with AHA demonstrate synergistic activity and prevent catheter encrustation in an in vitro model for CAUTI. Prophylactic use of urease dampening metabolites with AHA could improve the efficacy of antimicrobial treatment against catheter biofilms.

Catheter-associated urinary tract infections (CAUTIs) are the most common hospital acquired infections in the United States[1]. Each year more than 13,000 US healthcare-associated deaths are attributed to UTIs and ~75% are associated with a urinary catheter[2]. Between 15–25% of hospitalized patients receive urinary catheters during their hospital stay[2].

Urinary catheters facilitate bacterial colonization as they compromise many natural defenses against infection[3]. For example, indwelling catheters eliminate normal micturition and cause retention of 10–100 mL of urine within the bladder, providing a reservoir for bacteria to replicate[3–5]. Insertion and prolonged urinary catheterization also result in repeated mucosal and submucosal tears and

[1]Department of Microbiology and Immunology, Jacobs School of Medicine and Biomedical Sciences, State University of New York at Buffalo, Buffalo, NY, USA. [2]Department of Chemistry, University at Buffalo, The State University of New York, Buffalo, NY, USA. [3]Research and Education in Energy, Environment and Water (RENEW), University at Buffalo, The State University of New York, Buffalo, NY, USA. [4]Department of Chemistry, Wake Forest University, Winston Salem, NC, USA. [5]Laboratory for Forensic Odontology Research, School of Dental Medicine, SUNY at Buffalo, B1 Squire Hall, S. Campus, Buffalo, NY, USA. [6]Present address: Laboratory of Inorganic and Nuclear Chemistry, Wadsworth Center, New York State Department of Health, Albany, NY, USA. ✉e-mail: chelsiea@buffalo.edu

inflammation, which leads to deposition of host factors such as fibrinogen that can promote adherence of uropathogens (eg. *Enterococcus faecalis*, *Staphylococcus aureus*, and *Candida albicans*)[4,6,7]. As a result, bacteriuria incidence increases 3–7% each day an indwelling urinary catheter is in place such that nearly all long-term catheterized individuals will experience continuous bacteriuria and at least one episode of CAUTI[7–9]. CAUTI is also a leading cause of secondary nosocomial bloodstream infections with an associated mortality of approximately 10%[9].

The Gram-negative bacterium *Proteus mirabilis* is a predominant cause of both CAUTI and secondary bloodstream infection, particularly in patients with long-term indwelling catheters[3,8,10,11]. *P. mirabilis* poses a significant challenge due to intrinsic tetracycline and polymyxin resistance and acquired resistance to aminoglycosides and fluoroquinolones[12]. One of the most well-studied virulence factors of *P. mirabilis* is its cytoplasmic urease enzyme, a nickel-metalloenzyme which hydrolyzes the urea present in urine to ammonia and carbon dioxide causing a rapid increase in urine pH, ion precipitation, crystal formation, and urinary stones[13]. Urease activity therefore facilitates catheter encrustation and formation of antibiotic-recalcitrant crystalline biofilms, and increases the risk of bacteremia, sepsis and death[3,14–16]. The only direct urease inhibitor approved by the Food and Drug Administration (FDA), acetohydroxamic acid (AHA), showed efficacy in preventing urinary stone formation in clinical trials[17]. However, AHA is known to induce severe side effects including teratogenesis, hemolytic anemia, psychoneurological and muscular symptoms that limit its clinical use[13,17]. As a result, there is a surge of research into alternative strategies to inhibit bacterial urease with less severe toxicological profiles.

The most common drug discovery approaches for urease inhibition focus on small molecules targeting the enzyme active site and efficacy is screened using purified Jack Bean Urease (JBU). Recent urease inhibitors act in at least one of the following three ways[1]: interference with the binding of the urease enzyme native substrate[2]; binding to or inhibiting the incorporation of active site nickel ions; or[3] interacting with the moveable flap that covers its active site[13,18,19]. However, inhibitors that are highly effective against the purified urease enzyme may fail to provide efficacy against intact bacteria due to issues with crossing the outer membrane of Gram-negative species as well as the cell wall. Leveraging biological systems to guide the exploration of novel inhibitors therefore offers a promising alternative to expand on traditional approaches.

In patients with long-term indwelling catheters, bacteriuria and CAUTI are generally polymicrobial[7,8,10,20]. Importantly, we previously demonstrated that common constituents of the catheterized urinary tract modulate *P. mirabilis* urease activity, influencing tissue damage, urinary stone formation, and incidence of bacteremia in mouse models of UTI and CAUTI[16,21–23]. Specifically, *Enterococcus faecalis* and *Providencia stuartii* enhance *P. mirabilis* urease activity and increase infection severity while *Morganella morganii* dampens *P. mirabilis* urease activity and reduces infection severity[21]. Furthermore, the dampening effect of *M. morganii* was dominant over the enhancing effect of *E. faecalis*[21].

In this work we perform a global untargeted metabolomics analysis of six urease modulatory bacterial species to identify microbial-derived urease-dampening metabolites. We demonstrate whether these urease dampening metabolites modulate activity through direct interaction with the urease enzyme and conduct pre-clinical assessment of their therapeutic potential to prevent catheter encrustation.

## Results

### Co-colonizing species modulate *P. mirabilis* urease activity via secreted small molecules

Several uropathogens secrete factors that modulate *P. mirabilis* urease activity (Supplementary Fig. 1)[16,21]. We therefore sought to further

characterize the urease-modulating factors by subjecting cell-free bacterial supernatants to a series of treatments prior to inoculation with live, whole-cell *P. mirabilis*: 3 kDa size-exclusion filtration and 1) boiling for 10 min, 2) five freeze-thaw cycles, 3) metal chelation (Chelex), and 4) supplementation with excess nickel (Fig. 1A, B). Compared to *P. mirabilis* incubated in an autologous supernatant, *P. mirabilis* incubated in *E. faecalis* supernatant exhibited enhanced urease activity (higher average AUC), whereas *P. mirabilis* incubated in *M. morganii* supernatant exhibited decreased urease activity (lower average AUC). These urease-modulating effects remained following all treatments (Fig. 1B), indicating that the urease-modulatory factors are likely to be heat-stable molecules that do not require metal cofactors or act as metallophores for the supply or sequestration of nickel from the urease apoenzyme. Since the 3kD filtered cell free supernatants of *M. morganii* and *E. faecalis* maintained their urease dampening and enhancing phenotypes respectively, the secreted urease modulating factors are also likely to be small molecules or small secreted proteins that are less than 30 amino acids. Given the narrow size range for proteomics and that the secreted factors were heat stable (Fig. 1B), we decided to pursue an untargeted metabolomics approach to screen for urease modulating metabolites.

### Seven microbial-derived metabolites reproducibly dampen *P. mirabilis* urease activity

To identify microbial-derived small molecules that dampen *P. mirabilis* urease activity, we utilized Metabolon Inc's comprehensive Precision Metabolomics™ LC-MS platform for untargeted global metabolomics. Cell-free supernatants were generated from two strains of *P. mirabilis* (controls), four urease-enhancing strains, and seven urease-dampening strains after a 90-minute incubation in sterile 0.9% saline (Supplementary Data 1, Supplementary Fig. 1). Ultrahigh performance liquid chromatography-tandem mass spectroscopy detected a total of 307 metabolites from the cell-free saline supernatants: 262 named metabolites and 45 that could not be assigned a specific chemical structure. Analysis of variance (ANOVA) contrasts were used to identify metabolites whose mean relative abundance differed significantly between groups (i.e., control strains, enhancing strains, and dampening strains). Thirty-four metabolites were prioritized due to a $\geq 5$-fold increase in relative abundance in supernatants from dampening species compared to control or enhancing species supernatants ($p < 0.05$ using ANOVA corrected for multiple comparisons) (Supplementary Data 2).

Four of the candidate urease-dampening metabolites reproducibly decreased *P. mirabilis* urease activity in a dose dependent manner: histamine dihydrochloride (CAS 51-45-6), leucylglycine (CAS 686-50-0), phenylpyruvate (CAS 156-06-9), and imidazole lactate (CAS 14403-45-3) (Fig. 1C–I, and Supplementary Fig. 2). We also screened 11 additional metabolites present in supernatants of the dampening strains that shared functional groups or biosynthetic pathways with the verified urease-dampening metabolites, through which we identified two additional hits: imidazole (CAS 288-32-4) and 4-imidazole acetate (CAS 3251-69-2) (Supplementary Data 3, Fig. 1J, K, and Supplementary Fig. 2).

For added rigor, multiple formulations and lots were tested for five of the seven verified dampening compounds (histamine, leucylglycine, phenylpyruvate, 4-imidzaole acetate, and L-imidazole lactate). While most exhibited similar activities, two different lots of imidazole lactate purchased from the same manufacturer showed very different activity profiles as well as different physical appearances (Sup. Figure 3). To confirm chemical identities, both lot numbers were analyzed by Liquid chromatography high-resolution mass spectrometry (LC-HRMS), proton nuclear magnetic resonance (1H-NMR) spectroscopy, and polarimetry (see details in Supplementary Methods). These analyses support that both lots are imidazole lactate (Supplementary Fig. 3C, D, and 4A, B) and the two

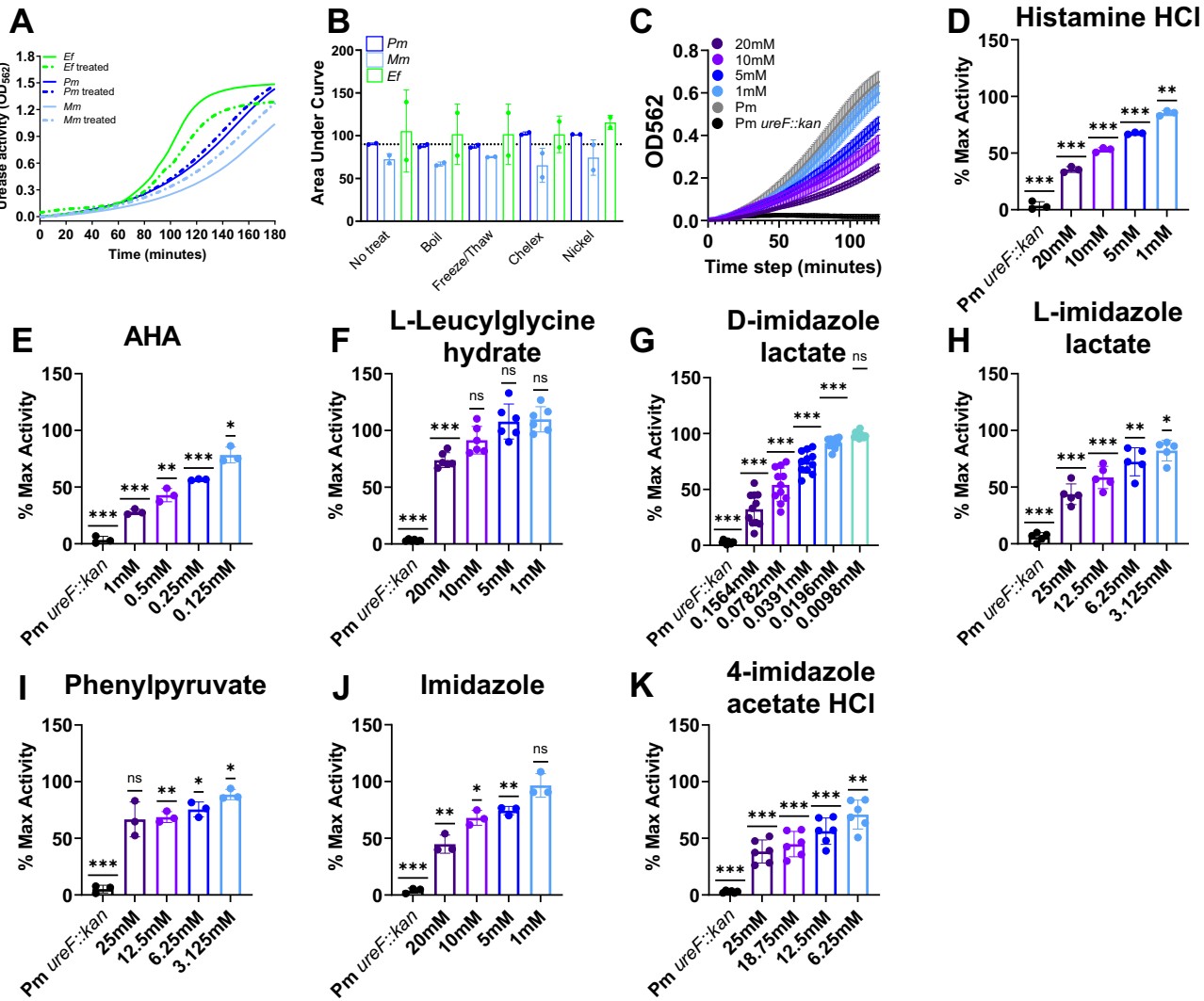

**Fig. 1 | Microbial metabolites dampen *P. mirabilis* urease activity. A** Urease activity of *P. mirabilis* (Pm) when incubated in cell-free supernatants from Pm HI4320 (control), *E. faecalis* 3143 (Ef), or *M. morganii* TA43 (Mm) with (solid line) or without (dashed line) 3 kDa filtration. Assays were conducted over 180 minutes at 37 °C with shaking and phenol red absorbance (OD562) was measured every 60 seconds. **B** 3kD filtered supernatants were subjected to the indicated treatments prior to assessing Pm urease activity. Activity from each 180-minute assay was expressed as Area Under the Curve (AUC). Error bars represent mean ± SD from 2 technical replicates in a single biologic experiment (**C**) Representative dose-response curve of Pm urease activity when incubated in a candidate dampening compound (histamine), with a urease mutant (Pm ureF::kan) as a negative control. **D–K** Urease activity of Pm incubated in candidate urease-dampening metabolites or the acetohydroxamic acid (**E**) positive control in potassium phosphate buffer. For plots (**D–K**) Activity is expressed as percent of maximum activity (% Max Activity) relative to untreated Pm and analyzed by two-tailed one sample t-test to a hypothetical value of 100%. All error bars represent mean ± SD from at least three independent experiments (Histamine *n* = 3, Leucylglycine *n* = 6, D-Imidazole lactate *n* = 11, L-Imidazole lactate *n* = 5, Phenylpyruvate *n* = 3, Imidazole *n* = 3, 4-Imidazole acetate *n* = 6, Acetohydroxamic acid (AHA, *n* = 3), with three technical replicates each. ns, not significant (*P* > 0.05); * *P* < 0.033; ** *P* < 0.002; *** *P* < 0.001.

imidazole lactate forms are enantiomers likely resulting in differing urease modulation activity; where D-imidazole lactate had a more potent activity profile compared to the L-imidazole lactate (Supplementary Fig. 3).

**Histamine, Leucylglycine, D-imidazole lactate, L-imidazole lactate, imidazole and 4-imidazole acetate dampen urease activity in cell free *P. mirabilis* extracts**

To begin examining mechanism of action, we first sought to determine whether the impact of urease-dampening metabolites on live, whole-cell *P. mirabilis* could be due to growth perturbation (Supplementary Fig. 5A, B). Growth in LB broth was only perturbed by D-imidazole lactate (Supplementary Fig. 5B, C), which also exhibited robust anti-microbial activity during incubation in potassium phosphate and less-pronounced activity in human urine (Supplementary Fig. 5C–E). However, it is notable that sub-inhibitory concentrations of

D-imidazole lactate (≤ 0.0391 mM) still decrease *P. mirabilis* urease activity, indicating additional mechanism(s) of action.

To assess whether dampening metabolites directly interact with the urease enzyme, we repeated the urease activity assay with cell free *P. mirabilis* extracts instead of whole-cell *P. mirabilis*. The FDA-approved urease inhibitor acetohydroxamic acid (AHA) was used as a positive control for all experiments, as AHA is a urea analog that directly interacts with the urease active site. Histamine, leucylglycine, D-imidazole lactate, L-imidazole lactate, imidazole and 4-imidazole acetate were all capable of at least partially inhibiting the urease activity of cell free *P. mirabilis* extracts while phenylpyruvate did not, suggesting a mechanism of action independent of the catalytically active urease enzyme for phenylpyruvate (Fig. 2). Analysis of Michaelis–Menten parameters in the presence and absence of urease dampening metabolites was conducted to indicate the type of inhibition (Supplementary Fig. 7–12). The addition of either histamine,

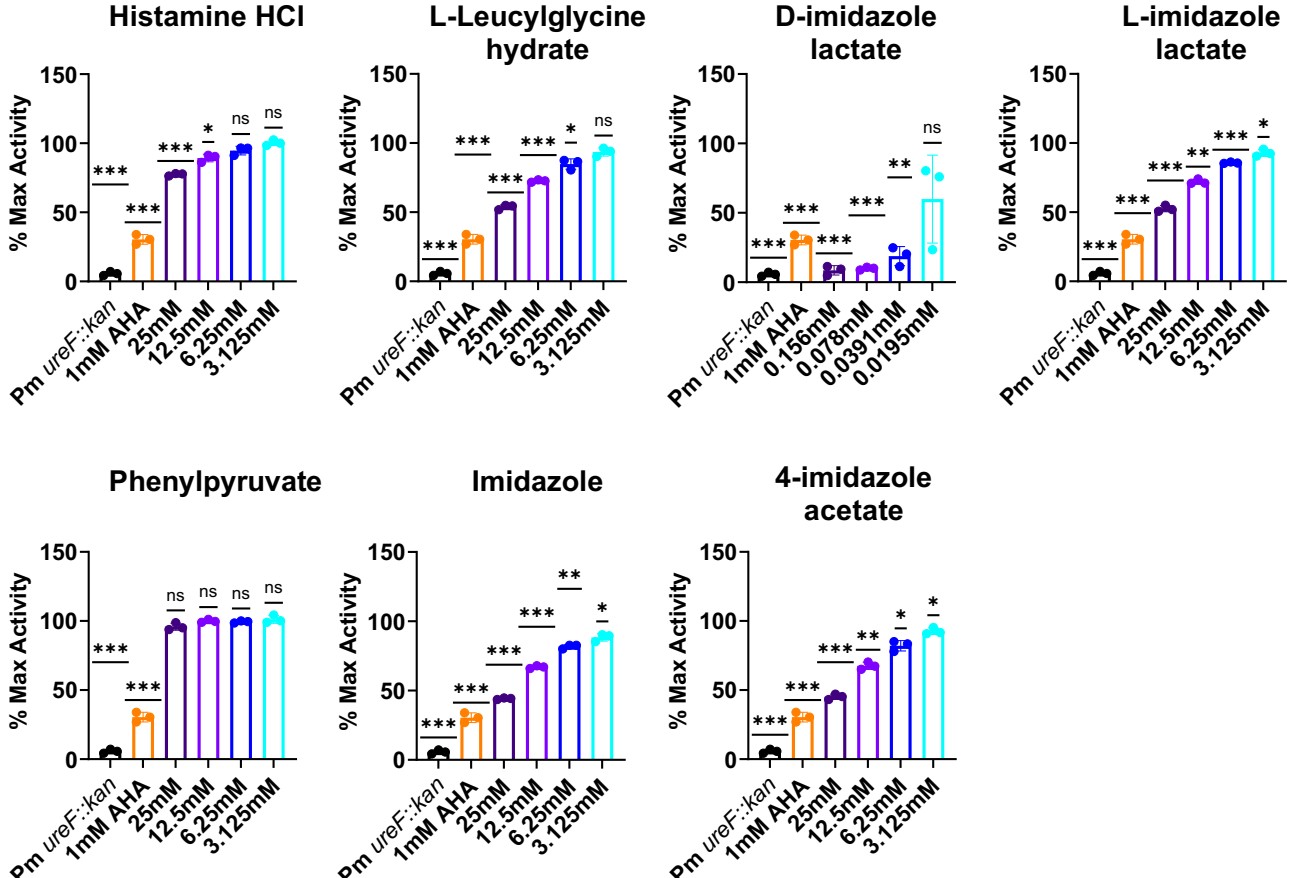

**Fig. 2 | Microbial metabolites dampen urease activity in cell free extracts of *P. mirabilis*.** Urease activity of *P. mirabilis* HI4320 cell free extracts when incubated in a candidate dampening metabolites or the 1 mM acetohydroxamic acid (AHA) positive control. A urease mutant lysate (Pm ureF::kan lys) serves as a negative control. Activity is expressed as percent of maximum activity (% Max Activity) relative to untreated Pm cell free extracts and analyzed by two-tailed one sample t-test to a hypothetical value of 100%. All error bars represent mean ± SD from three independent experiments, with three replicates each. *ns* not significant ($P > 0.05$); * $P < 0.033$; ** $P < 0.002$; *** $P < 0.001$.

leucylglycine, L-imidazole lactate, imidazole, or 4-imidazole acetate caused a decrease in the estimated Vmax, an increase in the estimated Km, and lack of intersection on either the x or y axis of the Lineweaver-Burke plot, suggesting that each of these metabolites act as mixed inhibitors (Supplementary Fig. 7–12). In contrast, the addition of D-imidazole lactate caused the estimated Vmax to decrease, the Km to remain roughly unchanged, and intersection on the x-axis of the Lineweaver-Burke plot, suggesting non-competitive inhibition (Supplementary Fig. 9). Analysis of Michaelis–Menten parameters was not possible for phenylpyruvate as this metabolite did not inhibit urease activity in cell free *P. mirabilis* extracts.

Inhibition of purified Jack Bean (*Cavavalia ensiformis*) urease enzyme (JBU) was also assessed as the enzyme active site is 100% conserved between *P. mirabilis* and JBU[24,25]. JBU activity was inhibited by leucylglycine, D-imidazole lactate, L-imidazole lactate, imidazole and 4-imidazole acetate, further suggesting inhibition through direct interaction with either the enzyme itself or the enzyme-substrate complex (Supplementary Figs. 13, 14). Interestingly, histamine only significantly inhibited JBU activity when added at concentrations higher than what was required to inhibit whole cell and cell free extracts of *P. mirabilis* urease activity (> 20 mM), while phenylpyruvate did not inhibit JBU activity (Supplementary Figs. 13–15). These data support a mechanism of action independent of enzyme binding for phenylpyruvate while histamine may interact with a portion of the enzyme that is not conserved between JBU and *P. mirabilis*. Notably, since D-imidazole lactate inhibited urease activity in cell free extracts

of *P. mirabilis* and purified JBU and was also the only urease-dampening metabolite with antimicrobial activity, these data suggest that it is capable of dampening urease activity via multiple enzyme-dependent and independent mechanisms (Fig. 2, and Supplementary Fig. 5, 9, 13, 14).

Since our urease activity screens for whole-cell and cell free extracts of *P. mirabilis* and purified JBU are both pH-based assays, we wanted to confirm that decreased activity is directly due to urease modulation rather than any intrinsic buffering capacity of the metabolite being tested. We therefore performed sodium hydroxide titration experiments. Three metabolites (imidazole, 4-imidazole acetate, and L-imidazole lactate) showed a consistent reduction in absorbance compared to the control across all concentrations of NaOH (Fig. 3), suggesting that high concentrations of these three metabolites can provide some degree of added buffering capacity that could contribute to the observed urease activity dampening.

### D-imidazole lactate decreases urea-dependent induction of urease operon expression

In *P. mirabilis*, the expression of the urease operon is rapidly induced by the urea-dependent transcriptional activator UreR[3]. To determine if any of the metabolites dampen *P. mirabilis* urease activity by altering the pool of urease mRNA transcripts, we performed quantitative reverse transcription PCR (RT-qPCR) using primers for the urease structural genes (*ureABC*), accessory subunit genes involved in nickel incorporation (*ureEDF*), and a housekeeping gene (*rpoA*). RNA was

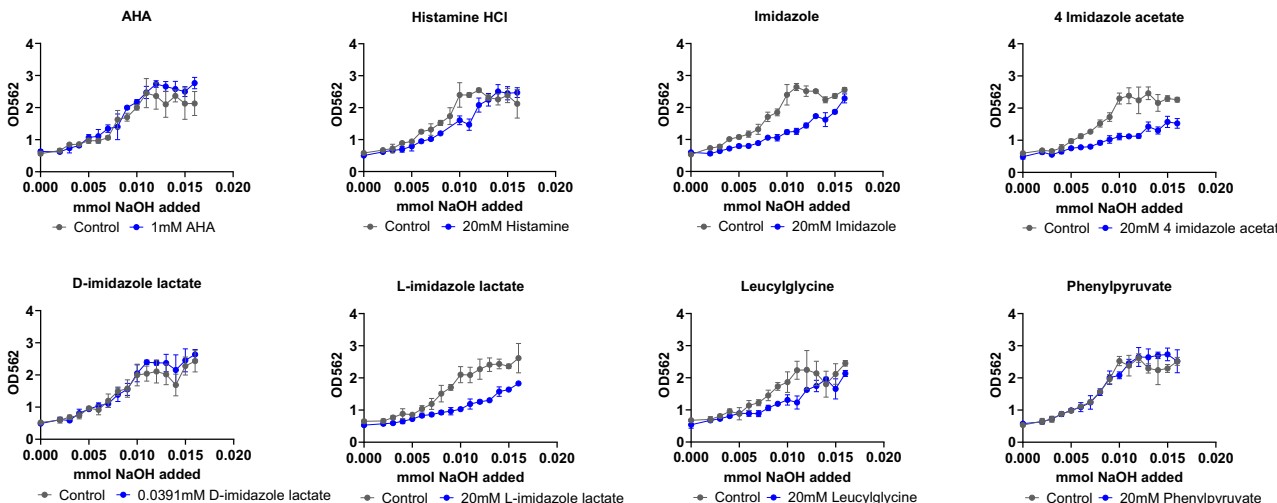

**Fig. 3 | Buffering capacity of urease-dampening metabolites.** The urease activity assay was modified such that instead of whole cell *P. mirabilis* a known amount of sodium hydroxide (NaOH) was added to potassium phosphate buffer supplemented with phenol red with or without (control) candidate dampening metabolite and absorbance (OD562) was measured. All data points and error bars represent mean ± SD respectively from three technical replicates.

isolated from mid-log phase *P. mirabilis* after a 15-minute incubation with or without 500 mM urea to induce expression of the urease operon in the presence or absence of each urease-dampening metabolite. Urea-mediated induction of gene expression was determined by comparing the level of each transcript in *P. mirabilis* incubated with urea in the presence or absence of each metabolite relative to expression in the no-urea control. D-imidazole lactate was the only metabolite that decreased the relative expression of urease mRNA transcripts (Fig. 4, and Supplementary Fig 16), indicating that it either inhibits urea-mediated activation of transcription or prevents accumulation of urease transcripts. Thus, D-imidazole lactate has at least three mechanisms of action depending on the concentration (non-competitive enzyme inhibition, reducing the pool of urease mRNA transcripts, and perturbing *P. mirabilis* growth). It is also notable that phenylpyruvate modestly increased *ureE* and *ureF* transcript levels (Fig. 4), despite dampening overall urease activity.

## Dampening metabolites have activity against other urease positive bacteria

To determine if the activity of urease-dampening metabolites was strain-specific against *P. mirabilis* HI4320, we tested dampening capacity against three other *P. mirabilis* clinical urinary isolates; one from an uncatheterized patient (HU1069)[26] and two from different participants in a recent prospective cohort study of long-term catheterized nursing home residents (104V0 and 106V15, whose urease operon have > 99% amino acid identity with *P. mirabilis* HI4320, Supplementary Fig. 18)[7]. All urease-dampening metabolites had similar activity profiles against all *P. mirabilis* strains except for histamine, which was slightly less potent against 104V0 and 106V15 than HI4320 and HU1069 (Supplementary Fig. 17).

In addition to *P. mirabilis*, several other common causes of persistent bacteriuria in catheterized patients also exhibit urease activity such as *Providencia stuartii*, *Morganella morganii*, *Klebsiella pneumonia*, *Pseudomonas aeruginosa*, and even Gram-positive species such as *Staphylococcus aureus*[7,8,10]. To further assess the broad translational potential of urease-dampening metabolites, activity was tested against whole cell *M. morganii*, *P. stuartii*, and methicillin-resistant *S. aureus* (MRSA). All compounds except phenylpyruvate significantly inhibited activity in *M. morganii* and *P. stuartii*, and D-imidazole lactate demonstrated the most potent urease inhibition (Fig. 5). For MRSA, all compounds were effective and phenylpyruvate demonstrated the most potent urease inhibition while D-imidazole lactate was the least

potent (Fig. 5). The effect of each metabolite on the growth and viability of these urease positive uropathogen species was also assessed. D-imidazole lactate was the only metabolite that perturbed growth, and it also demonstrated antimicrobial activity against each of the tested species (Supplementary Fig. 20). Interestingly, the antimicrobial activity of D-imidazole against UTI MRSA was modest, with only a one log reduction in CFUs after one h incubation in potassium phosphate buffer (Supplementary Fig 20). Thus, despite differences in mechanism of action, all seven urease-dampening metabolites are effective against multiple urease-positive species, including a Gram-positive pathogen.

## Histamine and 4-imidazole acetate synergize with AHA to reduce catheter encrustation in an in vitro CAUTI model

To assess the translational potential of these urease-dampening metabolites, we first verified that each candidate remained active in human urine and was non-cytotoxic. All seven urease-dampening metabolites reproducibly decreased *P. mirabilis* urease activity in a dose dependent manner, although less-pronounced, in human urine (Supplementary Fig. 21). Urease dampening metabolites were assessed for potential cytotoxic effects in monolayers of a human embryonic kidney cell line (HEK293) and a bladder epithelial cell line (T24) using an enzymatic cytotoxicity assay. None of the conditions tested demonstrated any reduction in viability of either cell line after a 24 h incubation period (Supplementary Fig. 22). These findings are encouraging for drug development and as they demonstrate a good starting point for further optimization.

Since AHA is a competitive inhibitor of the urease enzyme and all metabolites appear to act as either mixed or non-competitive inhibitors, we hypothesized that combinatorial use could reduce catheter encrustation and ultimately improve patient outcomes. Efficacy of all 28 combinations was compared to each metabolite administered alone using a combination index derived from the Loewe Additivity model[27]; 10 combinations exhibited significant synergistic dampening of *P. mirabilis* urease activity, while 4 combinations (AHA + leucylglycine, L-imidazole lactate + 4-imidazole acetate, leucylglycine + imidazole, and histamine + D-imidazole lactate) showed antagonistic activity (Fig. 6, and Supplementary Data 5). 

Histamine and 4-imidazole acetate were selected for further testing with AHA in a clinically-representative in vitro "bladder" model of catheter encrustation[28] as these two urease-dampening metabolites exhibited significant synergistic inhibition of *P. mirabilis* urease

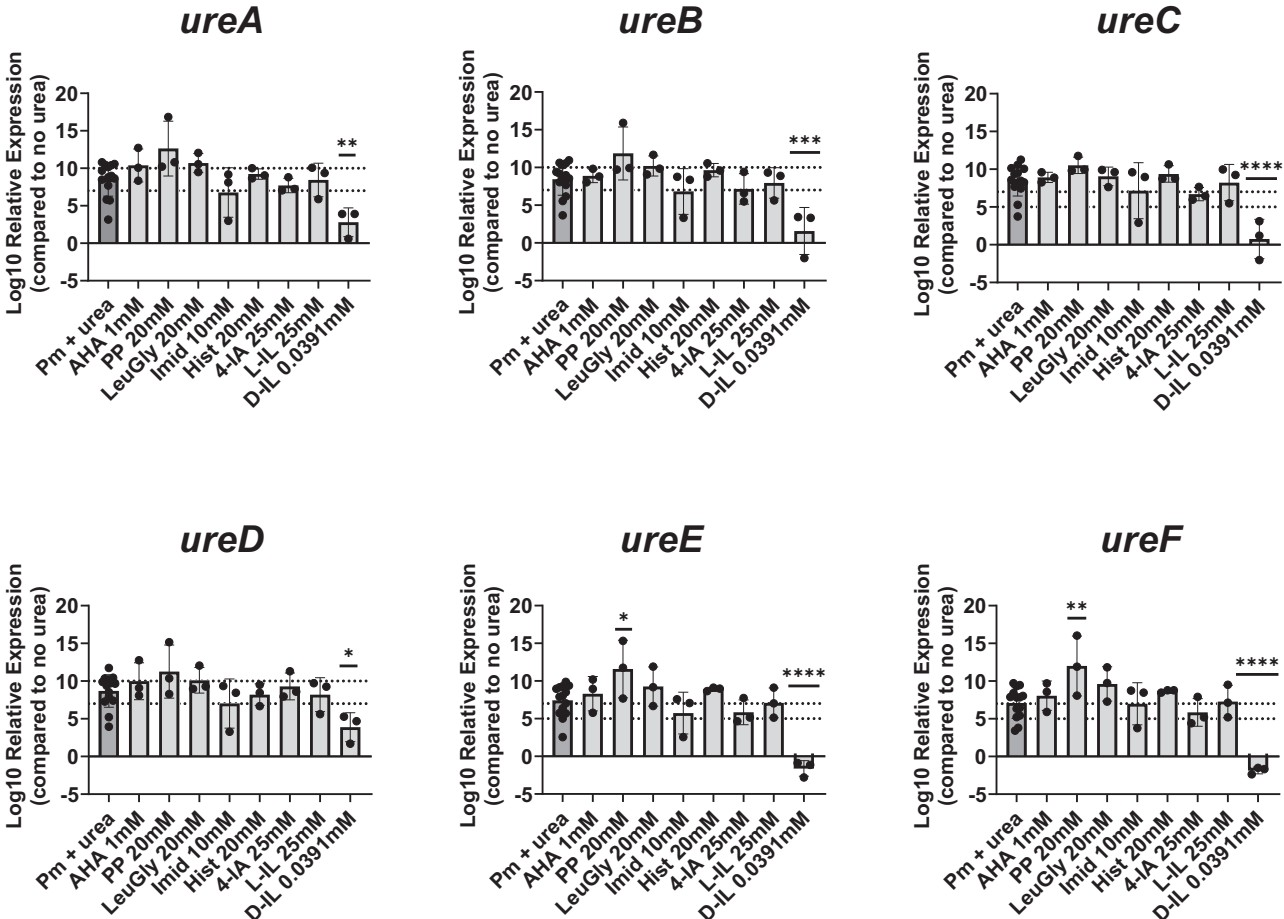

**Fig. 4 | D-imidazole lactate decreases expression of the *P. mirabilis* urease operon.** Expression level of urease structural genes (*ureABC*), accessory subunit genes involved in nickel incorporation (*ureEDF*), and a housekeeping gene (*rpoA*) were assessed by qRT PCR using RNA isolated from *P. mirabilis* after a 15 minute incubation in buffer alone, buffer with 500 mM urea, or buffer with urea and indicated dampening metabolites. Data were analyzed according to the Relative Quantification (RQ) method by Pfaffl et al. 2001. All error bars represent the mean ± SD of the Log10 RQ for RNA extracted from at least 3 independent experiments (Pm+urea *n* = 14 and all other conditions *n* = 3). Log10 transformed RQ of treated and untreated *P. mirabilis* was compared by ordinary One-Way ANOVA. \*\*\**P* < 0.0002, \*\**P* < 0.0021, \**P* < 0.0332. Acetohydroxamic acid *AHA*, histamine *Hist*, leucylglycine *LeuGly*, D-imidazole lactate *D-IL*, L-imidazole lactate *L-IL*, phenylpyruvate *PP*, imidazole (*Imid*), 4-imidazole acetate *4-IA*.

activity (CI < 1, Fig. 6) at concentrations <50 mM (Supplementary Data 5) and they exhibited the greatest reduction in the required dose of AHA when tested in AUM (Supplementary Fig. 23). Briefly, this model system consists of 500 mL jacketed glass vessel ("bladder") maintained at 37 °C by a circulating water bath. A size 14 F Foley catheter is inserted aseptically into the vessel through an opening at the base, the balloon is inflated as in a human patient to provide a seal, and the catheter is attached to a drainage bag (Fig. 7A). Since each "bladder" requires 2 liters of media every 24 h, these experiments were conducted using artificial urine media (AUM). The "bladder" is supplied with sterile AUM containing 500 mM urea at a physiologically relevant flow rate of 0.7–1.5 mL/min via peristaltic pump from a media reservoir, the "kidney" (Fig. 7A). An initial bacterial inoculum of $10^8$ CFU of *P. mirabilis* was added to the AUM in the "bladder" inner chamber to model robust bacteriuria, followed by initiation of flow (Fig. 7B)[24]. Urease-dampening metabolites and AHA were introduced into the system by direct dissolution into the AUM reservoirs, and the combinations tested were 1.25 mM AHA + 12.5 mM Histamine and 1.25 mM AHA + 12.5 mM 4-imidazole acetate as they reduced *P. mirabilis* urease activity by 90% in AUM with 500 mM urea in our standard urease activity assay (Supplementary Fig. 23A, B).

Since catheter encrustation is driven by urease-mediated pH increases and biofilm formation is enhanced by crystalline ion precipitation[3], the performance of urease-dampening metabolites was evaluated as a reduction in 1) AUM pH over time, and 2) catheter encrustation, measured as catheter crystalline biofilm formation and ion precipitation. *P. mirabilis* treated with urease-dampening metabolites was compared to untreated *P. mirabilis* and to *P. mirabilis ureF::kan*, which serves as a no-urease control to account for any potential impact of urease-dampening metabolites on bacterial viability and urease-independent biofilm production. To ensure that any observed effects on experimental outcomes were not driven by changes in bacterial viability, we also enumerated CFUs at regular intervals (0, 3, 6, 9, 12, and 24 h post inoculation) by collecting drained AUM through the catheter sampling port (Fig. 7B).

In the absence of a urease-dampening compound, the AUM collected from "bladders" inoculated with wild-*type P. mirabilis* reached a pH of ~8.0 within 3 h post inoculation (Fig. 7, and Supplementary Fig. 24). In contrast, AUM from "bladders" inoculated with the *P. mirabilis* urease mutant remained at ~6.0. All inhibitors (AHA, histamine and 4-imidazole acetate) perturbed urease-mediated pH increase in a dose-dependent manner when tested individually (Supplementary Fig. 24A–C), and this was not driven by changes in *P. mirabilis* viability as there were no meaningful differences between CFUs over time (Supplementary Fig. 24C–F). The pH of AUM collected from "bladders"

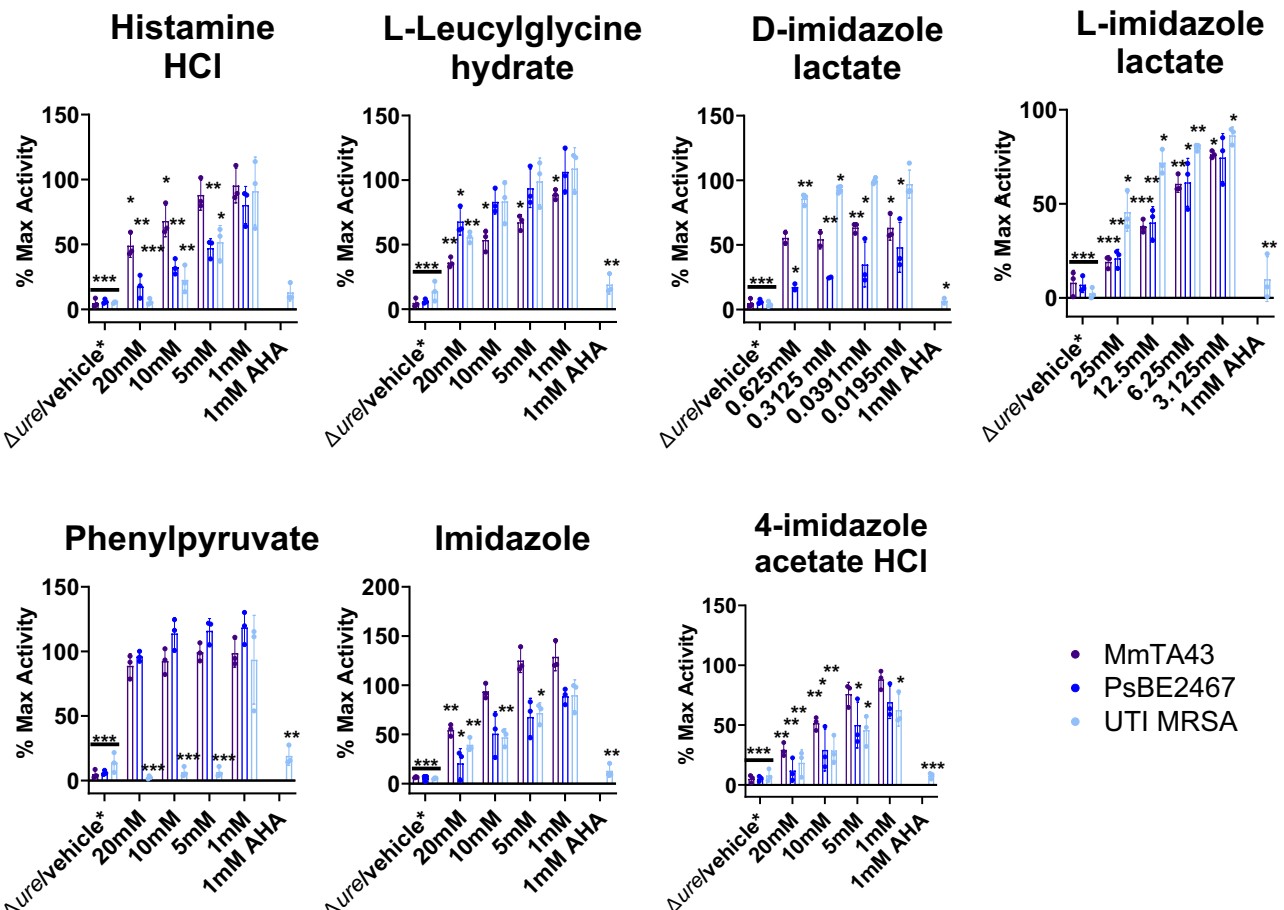

**Fig. 5 | Urease-dampening metabolites decrease urease activity of other bacteria.** *M. morganii* TA43 (MmTA43), *P. stuartii* BE2467 (PsBE2467) and UTI MRSA were incubated with the indicated dampening metabolites and urease activity was expressed as percent of maximum activity (% Max Activity) relative to untreated cultures. An isogenic *P. stuartii* urease mutant (PsBE2467 *Δure*) served as a negative control for PsBE2467, a vehicle (0.9% saline) control served as a negative control for MmTA43, and a *ureC* clean deletion mutant and transposon mutant in *ureC* (UTI MRSA *ureC::tn*) served as negative controls for UTI MRSA. Graphs show mean ± SD for 3 independent experiments with 3 technical replicates each. Statistical significance was determined by two-tailed One sample t test to a hypothetical value of 100%. ***$P < 0.001$, **$P < 0.002$, *$P < 0.033$.

treated with either 5 mM AHA, 25 mM histamine, or 25 mM 4-imidazole acetate alone never surpassed the nucleation point ( ~ pH 8) over the 24 h duration of the experiment while lower concentrations failed to restrict pH (Supplementary Fig. 24A–C), indicating that these concentrations are the lowest doses that might be effective at reducing catheter encrustation when used individually. Excitingly, lower doses of either histamine or 4-imidazole acetate (12.5 mM) synergized with a lower dose of AHA (1.25 mM) to significantly prevent the rise in pH ($p < 0.001$ and $p = 0.027$ respectively) over time without effecting *P. mirabilis* viability (Fig. 7C–F).

At 24 h post inoculation, crystalline biofilm biomass, biofilm CFUs and ion precipitation were examined across the catheter. High concentrations of 5 mM AHA, 25 mM histamine and 25 mM 4-imidazole acetate alone significantly reduced biofilm biomass compared to untreated *P. mirabilis* (Fig. 8A, Supplementary Fig. 25). The synergistic combinations of 1.25 mM AHA + 12.5 mM 4-imidazole acetate also significantly reduced biofilm biomass while 1.25 mM AHA + 12.5 mM histamine was not statistically significant ($p = 0.0722$, Fig. 8A). In contrast, 1.25 mM AHA, 12.5 mM histamine and 12.5 mM 4-imidazole acetate had no impact on crystalline biofilm formation (Supplementary Fig. 25). Changes in biofilm biomass were not driven by changes in viability as there were no significant differences between CFUs (Fig. 8B, and Supplementary Fig. 25).

Precipitation of urinary salts in alkalinized urine results in struvite (a conglomeration of bacteria, magnesium-ammonium-phosphate ($NH_4MgPO_4$) crystals and protein matrix) or carbonate apatite ($Ca_5(PO_4)_3(OH)$) crystallization[13,29]. Therefore, the amount of calcium (Ca), magnesium (Mg) and phosphorus (P) precipitation on catheter segments was measured via inductively coupled plasma optical emission spectrometry (ICP-OES). The high concentrations of 5 mM AHA, 25 mM histamine and 25 mM 4-imidazole acetate alone, as well as synergistic concentrations of AHA + histamine and AHA + 4-imidazole acetate significantly reduced calcium ion precipitation compared to untreated *P. mirabilis* (Fig. 8C). Magnesium and phosphorus precipitation were also reduced, although the reduction was not statistically significant (Fig. 8D, E, and Supplementary Data 6). Taken together, these data demonstrate that microbial-derived metabolites elicit a synergistic reduction in catheter encrustation when administered in combination with AHA by targeting urease-mediated ion precipitation.

The degree of encrustation on the 2 cm catheter eyelet segments was visualized via stereomicroscopy and scanning electron microscopy (Fig. 8F–I, left). No crystalline material was observed on the extraluminal surface of the catheter eyelet removed from the treated *P. mirabilis* and untreated *P. mirabilis ureF::kan* "bladders" when visualized via the stereomicroscope (Fig. 8G–I, left). In contrast, catheter eyelet segments from untreated *P. mirabilis* "bladders"

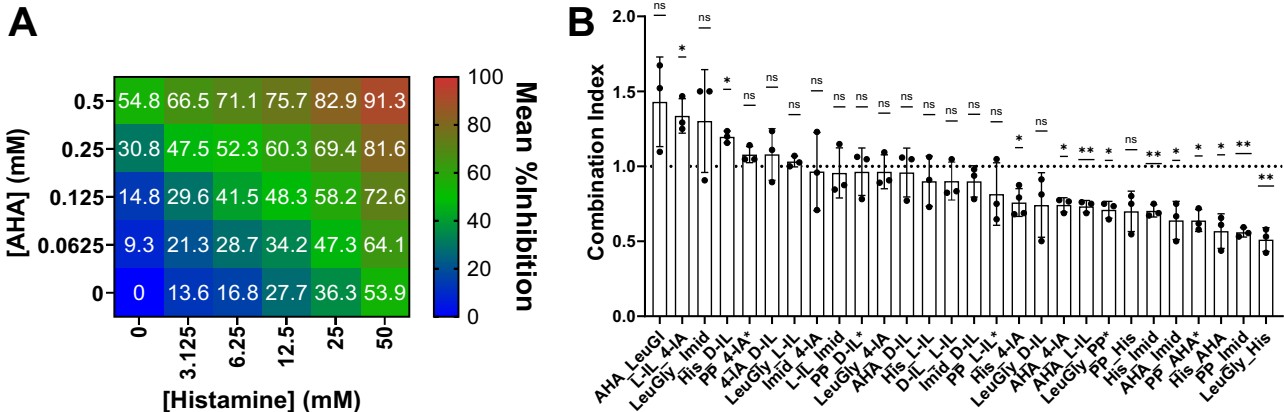

**Fig. 6 | Urease-dampening metabolites can synergize with each other or with AHA. A** Representative heat map of the mean percent inhibition of *P. mirabilis* urease activity when incubated in AHA and Histamine in combination and alone. **B** Combination index (CI) of all pairwise combinations of candidate urease-dampening metabolites with each other and with AHA. CI was calculated using the concentration of each urease-dampening metabolite that achieved the desired effect (40-80% inhibition of *P. mirabilis* urease activity) in combination and alone. Combinations for which the desired inhibition ( > 40%) could not be achieved were calculated using a lower percent inhibition and are indicated with an asterisk. Data represent mean ± SD for 3 independent experiments with at least 3 replicates each. Statistical significance was determined by two-tailed One sample t test to a hypothetical value of 1 (additive effect). ***$P < 0.001$, **$P < 0.002$, *$P < 0.01$. Acetohydroxamic acid *AHA*, histamine *His*, leucylglycine *LeuGly*, D-imidazole lactate *D-IL*, L-imidazole lactate *L-IL*, phenylpyruvate *PP*, imidazole(*Imid*, 4-imidazole acetate (*4-IA*).

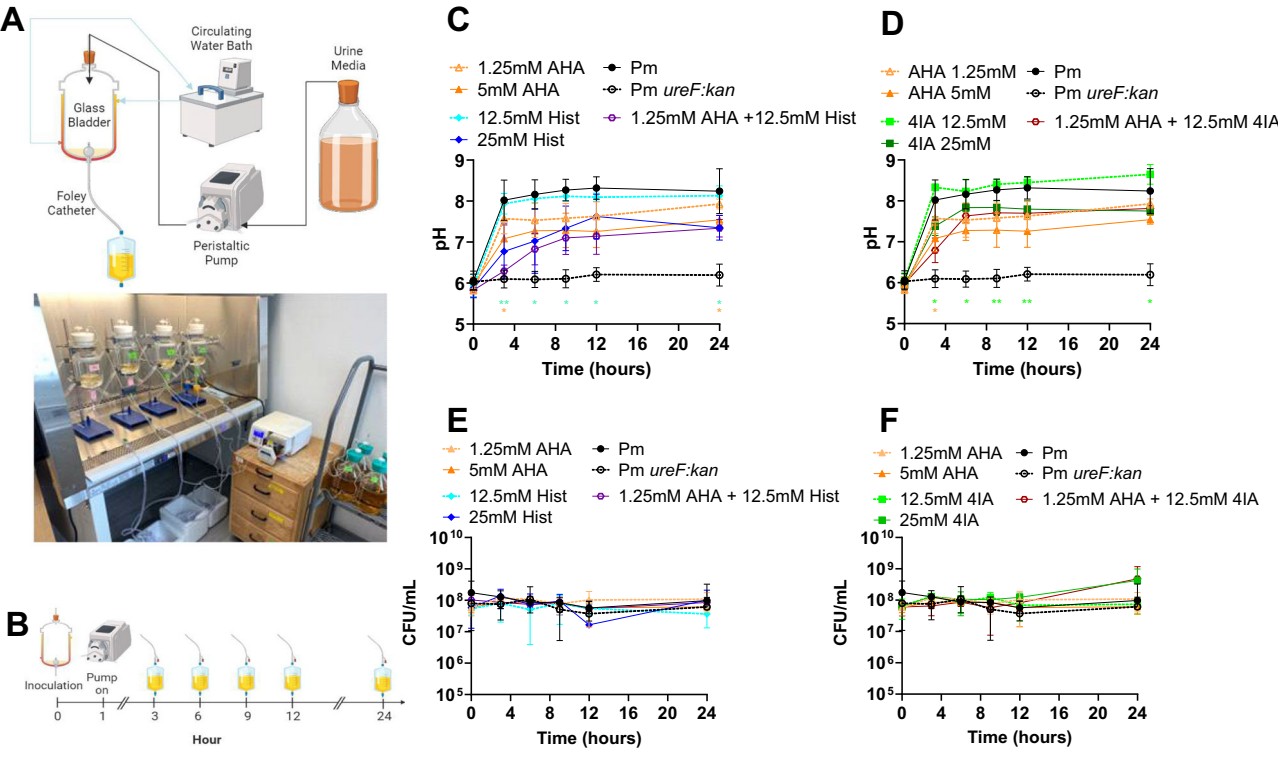

**Fig. 7 | Two urease-dampening metabolites synergize with AHA to prevent pH increases in an in vitro CAUTI model without altering bacterial viability. A** Schematic and image of the glass bladder model setup with five bladders running in tandem. Schematic adapted from Stickler et al.[55], and image created in BioRender (Guterman, B.https://BioRender.com/0d8si2e). **B** An initial bacterial inoculum of $10^8$ CFU of *P. mirabilis* or *P. mirabilis ureF::kan* was added to AUM in the "bladder" inner chamber, followed by initiation of flow and regular effluent collection over the 24 h study period (Created in BioRender. Guterman, B. (https://BioRender.com/kme7ioi). **C**, **D** pH and **E**, **F** CFUs of *P. mirabilis* were enumerated from the effluent collected through the catheter port for each bladder condition at 0, 3, 6, 9, 12, and 24 h post inoculation. Data represent mean ± SD for at least three independent glass bladder experiments (Pm $n = 18$, Pm *ureF::kan* $n = 18$, 1.25 mM AHA $n = 3$, 5 mM AHA $n = 3$, 12.5 mM Hist $n = 3$, 25 mM Hist $n = 3$, 1.25 mM AHA + 12.5 mM Hist $n = 3$, 12.5 mM 4IA $n = 3$, 25 mM 4IA $n = 3$, 1.25 mM AHA + 12.5 mM 4IA $n = 3$). Acetohydroxamic acid (AHA), histamine (Hist), 4-imidazole acetate (4-IA). Statistical significance of the combination treatment was determined by two-way ANOVA with multiple comparison corrections against the respective individual treatments. **$P < 0.01$, *$P < 0.05$.

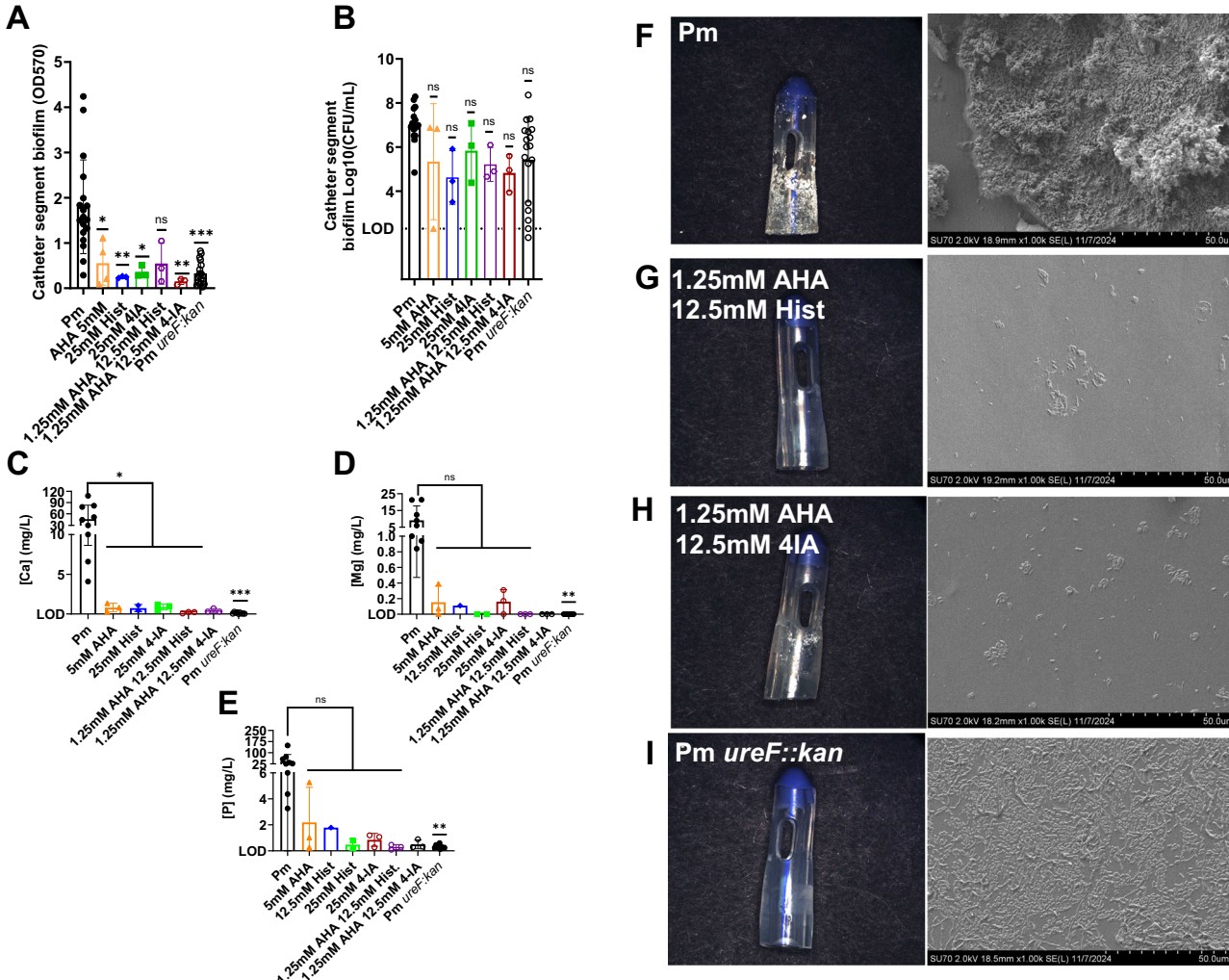

**Fig. 8 | Two urease-dampening metabolites synergize with AHA to prevent crystalline biofilm formation in an in vitro CAUTI model without altering bacterial viability.** Catheter encrustation was measured on 10 mm catheter segments 24 h post-inoculation by (**A**) crystal violet staining, **B** *P. mirabilis* CFUs, and **C–E** precipitation of calcium (Ca), magnesium (Mg), and phosphorous (P). Data in **A**, **B** (Pm *n* = 18, Pm *ureF::kan* *n* = 18, 5 mM AHA *n* = 3, 25 mM Hist *n* = 3, 1.25 mM AHA + 12.5 mM Hist *n* = 3, 25 mM 4IA *n* = 3, 1.25 mM AHA + 12.5 mM 4IA *n* = 3) and C-E (Pm *n* = 9, Pm *ureF::kan* *n* = 9, 5 mM AHA *n* = 3, 25 mM Hist *n* = 3, 1.25 mM

AHA + 12.5 mM Hist *n* = 3, 25 mM 4IA *n* = 3, 1.25 mM AHA + 12.5 mM 4IA *n* = 3) represent mean ± SD for at least three independent glass bladder experiments with three technical replicate catheter segments each, and were analyzed by One-way ANOVA with multiple comparisons. *ns* non-significant, * *p* < 0.05, ** *p* < 0.01, *** *p* < 0.001, **** *p* < 0.0001. **F–I** Stereo images (left) and scanning electron micrographs (right) of the 2 cm catheter eyelet segment 24 hours post-inoculation. Acetohydroxamic acid (AHA), histamine (Hist), 4-imidazole acetate (4-IA).

were clearly encrusted with sheet-like aggregates of large crystals in and around the eyelet (Fig. 8F, left). Scanning electron micrographs of the luminal surface at higher magnification revealed the presence of large numbers of bacilli on the surface of amorphous aggregates typical of apatite and struvite (Fig. 8F, right). Biofilm composition was further assessed using energy dispersive spectroscopy (EDS, Supplementary Fig. 26). Untreated *P. mirabilis* consistently showed elevated levels in elemental compositions of calcium (Ca), sodium (Na), oxygen (O), and phosphorus (P) in the biofilm matrix compared to *P. mirabilis ureF::kan*, confirming the increased apatite and struvite formations observed by stereomicroscopy and ICP-EOS (Fig. 8C–I, and Supplementary Data 6, Supplementary Fig 26).

### Histamine in combination with AHA increases efficacy of anti-microbials on catheter segment biofilms

Since 12.5 mM Histamine + 1.25 mM AHA reduced catheter encrustation in the in vitro CAUTI model without reducing the bacterial load within the catheter biofilm (Fig. 9A), we next wanted to assess whether

this treatment could increase the efficacy of antimicrobials. We therefore inoculated the in vitro "bladder" with *P. mirabilis*, established biofilms over a 24 h period with or without dampening metabolites, and then introduced 1 µg/mL of ceftriaxone (Fig. 9A). Efficacy was evaluated as a reduction in 1) AUM pH over time, 2) CFUs of *P. mirabilis* enumerated from the effluent AUM, and 3) the CFUs enumerated from the solubilized crystalline biofilm 48 h post-inoculation.

AUM collected from *P. mirabilis* "bladders" treated with the sub-therapeutic dose of 1ug/mL ceftriaxone had no impact on pH compared to the control "bladder" (Fig. 9B) but demonstrated a 3-log reduction in CFU/mL by the end of the experiment (Fig. 9C). Ceftriaxone coupled with 12.5 mM Histamine + 1.25 mM AHA caused a more pronounced reduction in pH (Fig. 9B) and a more rapid decrease in CFU than ceftriaxone alone (Fig. 9C). Importantly, coupling ceftriaxone treatment with the synergistic concentrations of histamine and AHA reduced the bacterial load within the catheter biofilms to a similar degree as ceftriaxone treatment of the *P. mirabilis ureF::Kan* biofilms (Fig. 9D). Thus, prophylactic use of histamine with AHA could

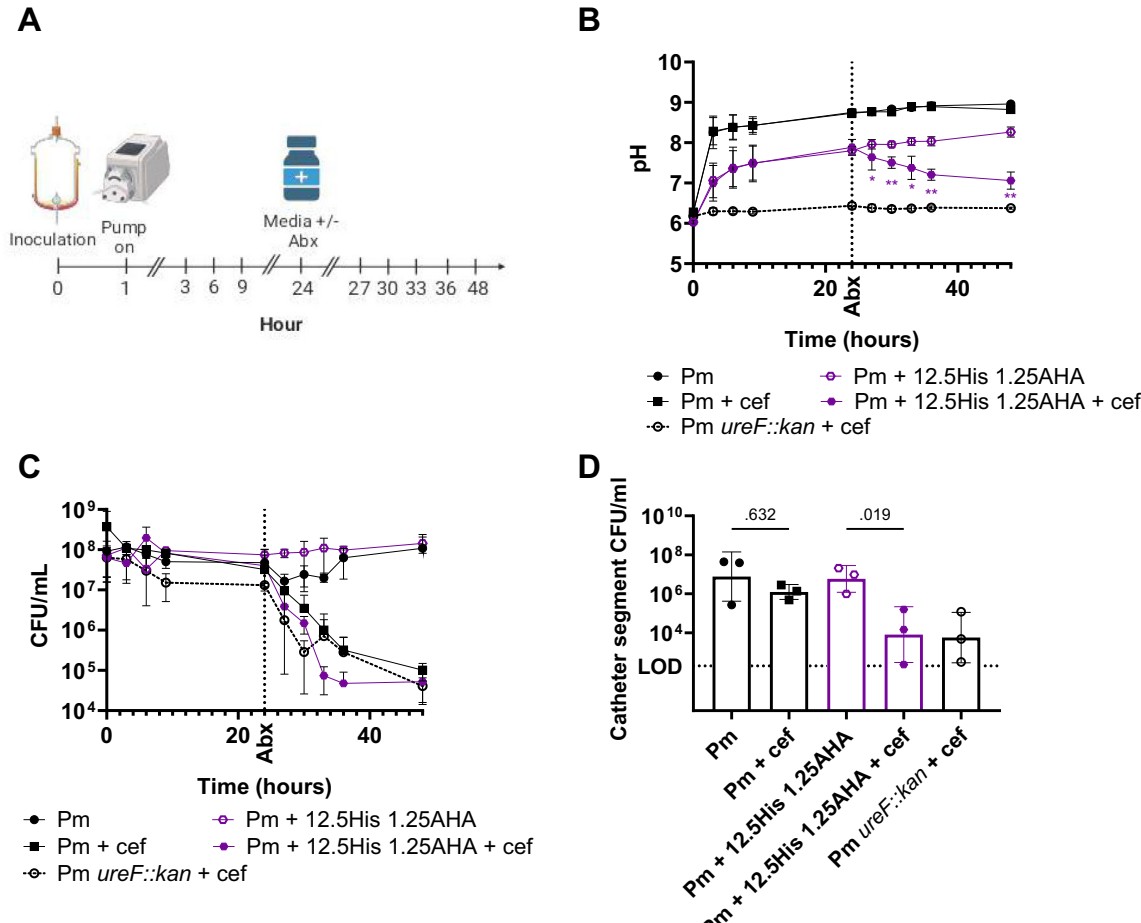

**Fig. 9 | Urease-dampening metabolites improve efficacy of antimicrobials in an in vitro CAUTI model. A** Schematic (created in BioRender: Guterman, https://BioRender.com/mbvw1in) for testing the ability of a synergistic histamine/AHA combination to improve efficacy of ceftriaxone (Abx). *P. mirabilis* catheter biofilms were established for 24 h with or without histamine and AHA prior to the addition of 1 μg/mL ceftriaxone (indicated with a dashed line). At 0, 3, 6, 9, 24, 27, 30, 33, 36 and 48 h post inoculation, pH (**B**) and CFUs (**C**) were enumerated from the effluent collected through the catheter port. Data represents mean ± SD for 3 independent experiments. Statistical significance of the synergistic histamine/AHA combination to improve efficacy of ceftriaxone was determined by two-way ANOVA with multiple comparison corrections against ceftriaxone alone. **$P < 0.01$, *$P < 0.05$. **D** CFUs enumerated from solubilized crystalline biofilm from 10 mm catheter segments 48 h post-inoculation. Data represent mean ± SD for at least three independent glass bladder experiments with three technical replicate catheter segments each, and were analyzed by One-way ANOVA with multiple comparisons.

improve the efficacy of antimicrobial treatment against catheter biofilms.

## Discussion

Bacterial urease is known to be important for pathogenesis and disease progression in a variety of contexts, including CAUTI[3,13,22]. There is growing appreciation for the fact that many infectious diseases are polymicrobial; however, there is still a paucity of studies addressing how these interactions contribute to disease severity. Previously, we demonstrated in patients with long-term indwelling catheters that bacteriuria and CAUTI are largely polymicrobial and that common constituents of the catheterized urinary tract modulate *P. mirabilis* urease activity, thereby influencing urinary stone formation and incidence of bacteremia[16,21]. Herein, we used untargeted global metabolomics to identify seven metabolites that dampen *P. mirabilis* urease activity (histamine, leucylglycine, D-imidazole lactate, L-imidazole lactate, phenylpyruvate, imidazole and 4-imidazole acetate) and assessed their translational potential in clinically relevant in vitro experiments. We further examined the synergistic effects of pairwise combinations of urease-dampening metabolites and identified combinations that synergize with the only FDA-approved urease inhibitor,

acetohydroxamic acid (AHA). AHA is widely recognized for its efficacy in reducing urease activity and mitigating the formation of urinary stones and catheter encrustation, but its clinical use is limited due to significant side effects, such as headaches, gastrointestinal disturbances, and even thromboembolic events[17]. Our data indicate that combinatorial treatment with microbial-derived urease-dampening metabolites reduces the concentration of AHA needed to prevent a urease-mediated rise in pH over time, and may therefore reduce clinical side effects while maintaining, or even enhancing, efficacy in preventing urinary stone formation and catheter encrustation. Furthermore, our data indicate that the combinatorial treatment of AHA with microbial-derived urease-dampening metabolites can improve the efficacy of antimicrobials in the management of CAUTI complicated by infection-induced catheter encrustation.

Considering that patients with long term catheterization typically exhibit polymicrobial bacteriuria and that several of the most common colonizing organisms in the catheterized urinary tract are urease positive[7], it is important to examine the impact of urease-dampening metabolites on other urinary tract pathogens. An advantage of our urease activity assay is that it utilizes whole bacterial cells, rather than isolated enzyme preparations. This design avoids complications

associated with metabolite import issues or concerns about ability to reach the target. All metabolites in this study, except for phenylpyruvate, significantly inhibited urease activity in all tested bacterial species. These data indicate that the urease-dampening compounds have a conserved target across both Gram-positive and Gram-negative bacteria. It was therefore surprising that not all of the dampening metabolites inhibited purified Jack Bean urease. One potential explanation is that the overall structure of JBU differs from the bacterial urease enzymes, despite having a conserved active site. For example, JBU is comprised of a single subunit compared to the more complex multi-subunit structures of bacterial ureases[30]. Therefore, our urease-dampening metabolites could still act directly on bacterial urease but outside of conserved portions of the enzyme. Interestingly, D-imidazole lactate was the most potent dampener of urease activity in the Gram-negative species but had minimal activity against MRSA. In contrast, phenylpyruvate potently inhibited MRSA urease activity but had only modest activity against *P. mirabilis* and no activity against *M. morganii, P. stuartii*, or JBU. Future work will focus on identifying the specific targets of each urease-dampening metabolite to guide structure activity relationship studies.

Of our seven urease-dampening metabolites, D-imidazole lactate was the only one with antimicrobial properties against clinical isolates of *P. mirabilis*, *M. morganii*, *P. stuartii* and MRSA. This aligns with the well-established role of imidazole as a key structural component in the development of antimicrobial agents. Imidazole, characterized by a five-membered heterocyclic ring, is commonly used in the pharmaceutical industry due to its ability to inhibit microbial processes, such as disrupting cell wall synthesis and integrity and interfering with membrane bound enzymes[31]. Despite the structural similarity, none of the other imidazole-containing compounds tested in this study exhibited antimicrobial activity. This suggests that while the imidazole ring is important for biological activity, the specific chemical environment surrounding the imidazole moiety, including side chains, chirality, or additional functional groups, plays a significant role in determining the antimicrobial efficacy of a given compound. In addition, D-imidazole lactate had less antimicrobial activity against MRSA compared to the Gram-negative species, suggesting possible differences in transport across the bacterial cell wall or target accessibility. Further testing is needed to elucidate the mechanism of D-imidazole lactate bactericidal activity such as assessing its impact on cell wall integrity and membrane permeability.

Notably, imidazole and its derivatives have also previously been reported to act as direct urease inhibitors[32]. For example, a series of oxadiazoles and thiadiazoles demonstrated competitive inhibition of Jack Bean, *Bacillus pasteurii* and *Proteus mirabilis* urease enzymes[30,33]. Omeprazole and lansoprazole inhibit *Helicobacter pylori* urease activity, and this was found to occur via covalent modification of cysteine residues at the active site that are likely conserved[30]. Interestingly, omeprazole and lansoprazole did not inhibit *P. mirabilis* urease activity, although the lack of activity may be due to the limited concentrations tested[34]. These findings suggest that the inhibitory effects of imidazole derivatives on urease activity may not be conserved across bacterial species and/or that the functional groups associated with the imidazole moiety play a more significant role in the mechanism of urease enzyme inhibition.

In addition to analysis of Michaelis–Menten parameters of cell free extracts of *P. mirabilis* in the presence of urease dampening metabolites, we also examined the impact of urease-dampening metabolites on expression of the urease operon. Of the seven urease-modulating metabolites, only D-imidazole lactate showed a significant reduction in expression of urease operon subunits in *P. mirabilis*. The reduction in both structural and accessory subunits of the urease operon by D-imidazole lactate suggests it may be interfering with the function or expression of the urea-inducible urease operon regulator, *ureR*. One other urease inhibitor, 2-Hydroxy-4-

methoxybenzaldehyde, was reported to reduce urease operon expression as well as overall urease activity, extracellular polysaccharide formation, and the formation of crystals by reducing the expression of *ureR* in addition to other virulence factors[35]. Future research is needed to determine how D-imidazole lactate interferes with urease operon expression as this may identify new targets for urease inhibition.

There are several limitations to this study that need to be addressed before application to human health. One limitation pertains to the millimolar concentrations used in the catheter encrustation model for histamine, 4-imidazole acetate, and AHA. A deeper understanding of the mechanisms through which these compounds modulate urease activity will facilitate modifications to enhance potency, making them more effective at clinically relevant concentrations. It is also important to note that the in vitro model for catheter encrustation does not incorporate host factors. Long term catheterization causes repeated damage to the bladder urothelial barrier and the associated persistent inflammation leads to the deposition of innate immune system host factors, such as fibrinogen, that can result in enhancement of bacterial colonization and biofilm formation[6]. In addition, we used artificial urine media (AUM) for the in vitro model. AUM has a greater buffering capacity than that of human urine, which may underestimate the potential of histamine and 4-imidazole acetate to maintain a pH below the nucleation point. In contrast, human urine has more proteins and metabolites that can potentially interfere with the effectiveness of histamine and 4-imidazole acetate to inhibit urease activity. Future studies should assess the effectiveness of metabolite combinations in "bladder" organoid models[36] or murine models of CAUTI[21] to examine potential effects on host tissue in human urine and further evaluate therapeutic potential.

We have determined that the concentrations used in our glass "bladder" system are not cytotoxic towards kidney and bladder epithelial cells over a 24 h incubation period, which is encouraging in drug development. However, the literature is divided on whether AHA is cytotoxic to mammalian cell lines. Our data aligns with the findings of Estrafeva et al. who reported that AHA did not significantly affect colorectal adenocarcinoma cells (Caco-2) cell viability up to 10 mM[37]. Similarly, Manoharan et al. showed 10 mM AHA had no cytotoxic effects on the human bladder epithelial cell line (BECs) 5637[38]. In contrast, Milo et al. found that 10 mM AHA caused a 30% decrease in viability of HaCaT cells[24] and Heylen et al. showed a 60-70% decrease in viability of human gastric cancer cells (SGC-7901), lymphoblasts (K562), and epithelial-like hepatocellular carcinoma cells (HepG2)[39]. The wide variability is likely due to differences in cell lines and methods for assessing cytotoxicity, but further testing is needed to verify the impact of the urease-dampening metabolites.

Histamine is a potent inflammatory mediator released from mast cells and has been implicated in the pathogenesis of various lower urinary tract disorders[24]. The histaminic receptor system is involved in modulation of bladder contraction and spontaneous activity, and administration of histamine was previously shown to induce a contractile response in isolated guinea pig and rabbit bladders as well as stimulation of histamine receptors in the lamina propria[40]. Thus, the effect of histamine on smooth muscle and bladder epithelial cells limits translational potential and necessitates further structural modification. It is also notable that 4-imidazole acetate has pharmacologic effects, particularly in the brain, as it is a structural analog of 4-aminobutyric acid (GABA). When administered into the lateral ventricle in cats, 4-imidazole acetate demonstrated a dose-dependent decrease in blood pressure. Intraperitoneal injections of 4-imidazole acetate produced sleep-like state and a decrease in body temperature in mice. Moreover, higher doses of 4-imidazole acetate caused seizures in rats four h post intraperitoneal administration[41]. Although these data are primarily in the central nervous systems of animal models, GABA receptors are found in wide range of peripheral tissues,

including the urinary bladder[42]. Therefore, future work will require modifications to histamine and to 4-imidazole acetate to inhibit their capacity to engage with histaminic and GABA receptors while retaining or even enhancing urease-dampening capacity.

Despite these limitations, our model demonstrates the potential to harness microbial-derived metabolites to reduce crystalline biofilm formation, catheter encrustation, and subsequent development of urinary stones. This approach represents a novel and highly target-specific therapeutic strategy that could be applied to combating ureolytic bacteria in many different settings. It is also notable that only one of the urease-dampening metabolites was antimicrobial, decreasing the risk of resistance. This approach opens new avenues for optimizing the therapeutic application of AHA, improving patient outcomes and minimizing adverse effects. Potential future applications could include a catheter flush solution or preventative catheter coating for patients with frequent blockage, urine stones, or persistent colonization by ureolytic bacteria. Ultimately, use of a cocktail of microbial-derived metabolites in combination with AHA would provide a novel approach for preventing UTI-associated bacteremia and sepsis in hospital and long term care settings.

## Methods

### Experimental design

The objective of this study was to identify microbial-derived metabolites that inhibit *P. mirabilis* urease activity, assess their potential mechanism of action of urease modulation, and conduct pre-clinical assessment of their therapeutic potential. Microbial-derived metabolites were identified using untargeted global metabolomics of cell-free supernatants from six urease-modulating bacterial species. An alkalimetric screen was used to measure urease activity in whole bacterial cells to identify metabolites that dampen *P. mirabilis* urease activity.

Representative in vitro models of the catheterized urinary tract, simulating a complete closed drainage system as used in clinical practice, were employed to evaluate the performance of the top urease-dampening metabolites, as detailed in Fig. 7A, B. This model system consists of jacketed glass vessel ("bladder") maintained at 37 °C by a circulating water bath. A Foley catheter is inserted aseptically into the vessel through an opening at the base, the balloon is inflated, and the catheter is attached to a catheter drainage bag (Fig. 7A). The "bladder" is supplied with sterile AUM at a physiologically relevant flow rate via peristaltic pump from a media reservoir, the "kidney" (Fig. 7A). An initial bacterial inoculum of $10^8$ CFU of *P. mirabilis* was added to the AUM in the "bladder" inner chamber to model robust bacteriuria, followed by initiation of flow (Fig. 7B). The performance of urease-dampening metabolites was evaluated as a reduction in 1) effluent AUM pH over the 24 h study period, and 2) crystalline biofilm formation and ion precipitation on catheter segments 24 h post inoculation. Urease-dampening metabolites and AHA were introduced into the system by direct dissolution into the AUM reservoirs. For each study, both positive (untreated *P. mirabilis*) and negative (untreated *P. mirabilis* urease negative mutant) controls were included. Effluent urine samples were collected at 0, 3, 6, 9, 12 and 24 h post inoculation. This preclinical model of catheter encrustation was modified to assess whether prophylactic administration of urease-dampening metabolites increases efficacy of antimicrobials against catheter biofilms, as outlined in Fig. 9A. Each experiment was performed in triplicate. Full details of experimental set up, sample collection, and processing are provided below and in Supplementary Materials.

### Bacterial strains and culture conditions

A complete list of bacterial strains used in this study is provided in Supplementary Data 1[43–47]. Bacteria were routinely cultured at 37 °C with aeration in 5 ml low-salt LB broth (10 g/liter tryptone, 5 g/liter yeast extract, 0.1 g/liter NaCl) or on low-salt LB agar (15 g/liter agar). *Enterococcus faecalis* and UTI MRSA isolates were cultured in brain

heart infusion (Difco) broth. Brain heart infusion broth was supplemented with 10 μg/ml Erythromycin for the transposon mutant in UTI MRSA *ureC* (*ureC::tn*).

### Urease activity assays

Urease activity was measured as described previously in triplicate with at least three biological replicates[21]. Briefly, bacteria were cultured for ~18 h in LB broth, concentrated to an $OD_{600}$ of 4, washed once in saline, and diluted (1:10 for *P. mirabilis* and 1:50 for *P. stuartii* and *M. morganii*) into candidate dampening metabolites in potassium phosphate buffer or human urine pH7 and supplemented with 500 mM urea and 0.01% phenol red. Bacterial suspensions were distributed into a 96-well plate, incubated at 37 °C with double-orbital shaking in a Synergy H1 plate reader (BioTek), and absorbance ($OD_{562}$) was measured every 60 seconds for 180 minutes. Isogenic urease mutants (Supplementary Data 1) served as negative controls for *P. mirabilis* and *P. stuartii* and a vehicle control served as a negative control for *M. morganii*.

To measure urease activity of the cell free extracts of *P. mirabilis*, *P. mirabilis* was cultured for ~18 h in LB broth and then sub cultured until it reached mid-log growth phase ($OD_{600}$ 0.2-0.5). Cultures were spun down to pellet at 10,000 rcf for 10 minutes and resuspended in saline to a final $OD_{600}$ of 1.0. Cultures were then diluted 1:10 in potassium phosphate buffer pH7 supplemented with 500 mM urea and incubated at 37 °C with shaking for 1 h to induce its urease enzyme production. The isogenic urease mutant Pm *ureF::kan* (Supplementary Data 1) served as negative control. Cultures then were spun down to pellet at 10,000 rcf for 10 minutes and resuspended in 2 mL saline. Cultures were subjected to four cycles of freezing in liquid nitrogen and thawing in a 37 °C water bath. Cultures were then probe sonicated (on ice) at 25% amplitude for 10 sec. Sonicated cultures were centrifuged at 18,900 Xg for 5 min and supernatants were transferred to a clean tube and stored on ice. To measure urease activity of cell free extracts, the previously described assay was modified such that 20ul of cell-free extract were distributed into a 96-well plate in place of the whole cell bacterial suspensions. Acetohydroxamic acid (AHA), was used as a positive control. Percent of maximum urease activity was calculated using Area Under the Curve (AUC) relative to untreated cell free extracts of *P. mirabilis*.

To measure Jack Bean urease activity, the previously described assay was modified such that 0.1 U (1U liberates 1 μmol of ammonia from urea per minute) of JBU (Sigma-Aldrich urease from *Canavalia ensiformis*) were distributed into a 96-well plate in place of the bacterial suspensions. Acetohydroxamic acid (AHA), was used as a positive control and a vehicle control was used as a negative control. Percent of maximum urease activity was calculated using Area Under the Curve (AUC) relative to untreated bacteria of interest or untreated JBU. Urease activity measurement in MRSA was adapted from a recently published semi quantitative assay (See Supplementary materials for details)[48].

### Kinetics of urease inhibition analysis

For the kinetic analysis of the urease reaction in the presence of urease dampening metabolites, urease activity assays were repeated with the *P. mirabilis* cell free extracts. Six different inhibitor concentrations that gave linear rates of reaction over a period of 120 min were tested for their effects on urease activity in the presence of urea concentrations ranging from 0 to 500 mM. Simple linear regression was performed on the absorbance ($OD_{562}$) over 120 minutes for each condition and plotted as Initial velocity (slope) vs urea concentration. These initial hyperbolic plots were converted to double-reciprocal Lineweaver-Burke plots, and the pattern of inhibition inferred from the point of intersection of the lines. Finally, the Ki for a particular compound was determined by plotting the Km/Vmax ratio (the slope of the Lineweaver-Burke plots) of each line as a function of the inhibitor

concentration as described by ref. [49]. The X-intercept corresponded to −Ki.

### Buffering capacity assay

Known concentrations of sodium hydroxide were added to each well of a 96 well plate in combination with candidate dampening metabolite of interest buffered 1:10 in potassium phosphate buffer pH7. 0.01% phenol red was added to each well, and absorbance (OD562) was measured in a Synergy H1 plate reader (BioTek).

### Generation and testing of cell free supernatants

Bacterial strains were cultured to mid-log phase ($OD_{600}$ of 0.2-0.5) in LB or BHI, centrifuged to pellet, and resuspended in 0.9% sterile saline at an $OD_{600}$ of 0.5. Resuspended cultures were incubated at 37 °C with aeration at 225 rpm for 90 min, centrifuged to pellet, and supernatants were passaged through a 0.22-μm-pore-size filter (Millipore Express® Plus Membrane) to generate cell-free supernatants. To further characterize urease-modulating factors, supernatants were subjected to 3 kDa size-exclusion filtration (Amicon ® Ultra Centrifugal filters) and/or treated as followed: 1) boiled for 10 min, 2) subjected to five cycles of freezing in a dry ice/ethanol bath and thawing in a 56 °C water bath, 3) treated with Chelex 100 to chelate metal ions, and 4) supplemented with 10 μM NiSO4.

### Sample preparation for Global Untargeted Metabolomics

Cell-free supernatants were generated as above from bacterial strains of interest (Supplementary Data 1). Suspensions were then centrifuged to pellet, filter-sterilized using a 0.22um filter, passaged through a 3 kDa size-exclusion filter (Amicon® Ultra Centrifugal Filter), and stored at -20 °C until use.

Saved supernatants from three independent replicates of each strain were sent to Metabolon Inc for global untargeted metabolomics analysis using the comprehensive Precision Metabolomics™ LI-MS platform[50,51]. MS2 spectra were acquired, and raw data from the main metabolomics experiments can be accessed in the MetaboLights database (https://www.ebi.ac.uk/metabolights/reviewer81d00c68-bf4d-4db6-ad69-412dfe00b64d). The measured intensities for each metabolite is the calculated area under the curve for the peak associated with each metabolite. "Peak Area Data" is scaled to the median value for each metabolite (all samples are ranked low to high; the middle value is set to 1.0 and all other values are scaled accordingly). This does not change the distribution of the data other than setting all metabolites on a similar scale. The median-scaled data are referred to as "Batch norm data." Any missing values in the median-scaled data are then imputed with the minimum value detected across the dataset[52]. As the Metabolon platform is optimized for the broadest metabolite coverage, the limit of detection (LOD) for any given compound is not the same and not necessarily optimized. Therefore, the percentage of samples in each bacterial cell free supernatant sample that have a metabolite detected is given as the "% Fill" in Supplementary Data 7. The "Batch norm data" were then natural log transformed for calculation of fold change between groups as well as statistical analysis to identify biochemicals that were significantly over- or under-represented in a given group compared to the control samples by one-way ANOVA with Tukey's test for multiple comparison (q=false discovery rate). "Fold change" was calculated as the ratio of the mean scaled intensity for a given metabolite between two experimental groups, using the batch-normalized imputed values (Supplementary Data 7). All reported biochemicals in Supplementary Data 7 that are given without an asterix (*) are considered Tier 1 identifications according to the Metabolomics Standards Initiative. Metabolites reported with either 1 (*) or 2 (**) asterisks do not officially meet the Tier 1 requirements, but do not drop to the Tier 2 level as there is sufficient other information to provide confidence in the identity of the biochemical. Therefore, they are designated as 'Not Tier 1' identifications.

Candidate urease-dampening metabolites were prioritized if they exhibited a fold change ≥5 for relative abundance in supernatants from any of the urease-dampening strains compared to the relative abundance in supernatants from any of the control or enhancing strains and the fold change was statistically significant ($p < 0.05$) with a false discovery rate (q) less than 0.1. Candidate dampening metabolites were excluded if the metabolite was not detected in any of the experimental or control groups (% Fill = 0).

### Generating stock solutions of urease-dampening metabolites

All metabolites included in the Metabolon standards library were reported with their associated CAS number. CAS numbers of prioritized candidate urease-dampening metabolites guided purchase, and all ordering and lot information is reported in Supplementary Data 2. Candidate urease-dampening metabolites and acetohydroxamic acid were dissolved in MiliQ water, titrated to a pH between 6 and 7, filter sterilized using a 0.2μm filter, aliquoted, and stored at -80 °C.

### Real-time quantitative PCR

*P. mirabilis* was cultured for ~18 h in LB broth and then sub cultured until it reached mid-log growth phase ($OD_{600}$ 0.2-0.5). Cultures were spun down to pellet at 10,000 rcf for 10 minutes and resuspended in saline to a final $OD_{600}$ of 1.0. Cultures were then diluted 1:10 into dampening metabolites buffered 1:10 in potassium phosphate buffer pH7 and supplemented with 500 mM urea. *P. mirabilis* in potassium phosphate buffer without urea served as a negative control. Cultures were incubated in potassium phosphate buffer with and without dampening metabolites at 37 °C with shaking for 15 minutes. RNA isolation, cDNA synthesis and RT qPCR were performed as described previously[53]. For analysis, *rpoA* was chosen as the reference gene for normalization between samples as it exhibited low variation in the presence and absence of urea (Supplementary Fig 16B-H). A RNA integrity analysis using agarose gel electrophoresis on extracted RNA was performed for all samples to confirm no change in transcript stability (Supplementary Fig 16A). The data were analyzed according to the Relative Quantification (RQ) method by Pfaffl et al. 2001[54] (See Supplementary Information).

### Synergy experiments

To assess the synergistic effects of dampening metabolites when used in combination with each other, *P. mirabilis* urease activity was measured as described above. Between 4-6 different concentrations for each dampening metabolite were tested both alone and in combination (See Fig. 6a for example). Concentrations were selected with the goal of achieving a percent inhibition of 40-85% (compared to untreated *P. mirabilis*). To account for variability between experiments, all conditions tested were performed on a single 96-well plate with three technical replicates for each condition. To compare between biologic replicates, percent inhibition relative to untreated *P. mirabilis* was calculated for each condition using the average area under the curve over 120-180 minutes (so that the positive control activity curve saturated without any increase in the negative control curve).

A combination index (CI)[27] was calculated using the concentrations of compounds and that achieved the desired effect (40-85% inhibition of *P. mirabilis* urease activity) in combination (combo) compared to when used alone.

$$CI = \frac{[Compound\ A_{combo}]}{[Compound\ A_{alone}]} + \frac{[Compound\ B_{combo}]}{[Compound\ B_{alone}]} \tag{1}$$

CI < 1 indicates a synergistic effect, C = 1 suggests an additive effect, and CI > 1 indicates an antagonist effect. Statistical analysis was

conducted using a one sample T test of the combination index compared to a hypothetical value of 1 (additive effect). The full data for all synergy testing can be found in Supplementary Data 5. Notably, not all conditions could satisfy the desired effect of %Inhibition > 40%. Therefore, combination indexes were calculated using a desired effect size less than 40% were indicated with an * in Fig. 6b and in Supplementary Data 5.

## In vitro "bladder" model

A previously described glass "bladder" model was used, in which a 500 mL water-jacketed glass vessel is maintained at 37 °C via a circulating water bath, a Foley catheter is inserted and inflated, and artificial urine medium (AUM) is supplied into the "bladder" at a constant flow rate (0.75-1.0 mL/min) through a peristaltic pump[55,56]. For consistency with the in vitro urease activity assays and the physiologic range of urea in healthy human urine, AUM was supplemented with 500 mM urea[57]. Urease-dampening metabolites of interest were directly dissolved in AUM 500 mM urea, adjusted to a pH of 5.8, and filter sterilized via a 0.2 μm filter. "Bladders" were inoculated with $10^8$ CFUs of *P. mirabilis*. Samples were collected from the catheter port at 0, 3, 6, 9, 12, and 24 hours to monitor pH and bacterial CFUs. Catheters segments were processed for CFUs, crystal violet staining of biofilm biomass, and ion composition at 24 h. Full details are provided in the Supplementary Methods.

## Antibiotic testing in the in vitro "bladder" model

The glass "bladder" system assembly, inoculation and sample collection were performed as described above. At 24 h post inoculation glass "bladders" were supplied with media containing the same metabolite (or AUM only control) with or without 1 μg/mL ceftriaxone sodium salt hydrate (Cayman Chemical Company, Michigan, USA). Samples were then collected from the catheter port at 27, 30, 33, 36 and 48 h to monitor pH and bacterial CFUs. After 48 h, the catheters were carefully removed, sectioned and assessed for bacterial viability as detailed in Supplementary Methods.

## Cytotoxicity assay

HEK293 (American Type Culture Collection) and T24 (ATCC HTB-4) cell lines were cultured for cytotoxicity evaluation as previously described (See Supplementary methods for details)[21]. Urease dampening metabolites were directly dissolved into the respective cell culture media, titrated to pH between 7.2-7.4, filter sterilized, aliquoted, and stored at -20 °C until use. After 24 h incubation, confluent monolayers were treated with fresh media supplemented with the urease-dampening metabolites to assess cytotoxicity. After a 24 h incubation at 37 °C and 5% $CO_2$, the amount of LDH released into the supernatant was measured using an LDH cytotoxicity assay kit (Cayman Chemical). Treatment with 10% Triton X-100 for the 24 h incubation was used as a positive control for maximum lysis, and treatment with cell culture media alone was used as a negative control. The level of cell lysis under each treatment condition is expressed relative to the level of spontaneous release (cell culture media alone).

## Statistics & reproducibility

Each experiment was performed in triplicate. Since experiments were in vitro and under controlled conditions, no statistical method was used to predetermine sample size and the experiments were not randomized. No data were excluded from the analyses. The Investigators were not blinded to experimental conditions and outcome assessment. Significance was assessed using one way and two-way analysis of variance (ANOVA) or one sample t-test to a hypothetical value as indicated in the figure legends. All P values are two tailed at a 95% confidence interval. All analyses were performed using Prism, version 10.50 for Windows 64-bit (GraphPad Software, San Diego, CA).

## Reporting summary

Further information on research design is available in the Nature Portfolio Reporting Summary linked to this article.

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

## Acknowledgements

We would like to thank Dr. T.R. and members of his laboratory, Dr. J.P., Dr. P.D., Dr. T.M., and Dr. E.W., for their helpful comments and critiques. This work was funded by the National Institute of Allergy and Infectious Disease under award R21 AI165979 (CEA) and the National Institute of Diabetes Digestive and Kidney Disease under award F30 DK136213 (LBG). The content is solely the responsibility of the authors and does not necessarily represent the official views of the National Institutes of Health.

## Author contributions

The microscopy and energy dispersive spectroscopy of catheter segments was performed with PJB at the University at Buffalo School of Dental Medicine Instrument Center. The elemental analysis by ICP-OES in this publication was performed by the G.L.D. and B.M.F. in the Donati Group in the Department of Chemistry, Wake Forest University. M.K.W., A.C.H., and D.S.A. confirmed chemical identities of D- and L-imidazole lactate. L.B.G. and C.E.A. designed experiments, analyzed data, and prepared the manuscript. L.B.G., M.M., B.C.H., A.L.B., B.S.L., B.F., M.K.W., A.C.H., G.L.D., B.M.F., P.J.B. and C.E.A. performed experiments. L.B.G. and M.K.W. prepared data visualizations. N.D. performed genome sequencing and alignment analysis. B.S.L. and A.L.B. provided project administration. All authors critically reviewed and revised the manuscript.

## Competing interests

The authors declare no competing interests
