## [Transparent Peer Review file · Nature Communications]

Harnessing microbial-derived metabolites in the urinary tract to prevent infection induced catheter encrustation

Corresponding Author: Dr Lauren Guterman

Version 0:

Reviewer comments:

Reviewer #1

(Remarks to the Author)

In this manuscript, the authors investigated potential alternatives to AHA for treating catheter-associated urinary tract infections (CAUTIs) caused by the urease activity from bacteria such as *Proteus mirabilis*. They started from the observation that several uropathogens are able to modulate *P. mirabilis* urease activity through secreted factors.

Using a metabolomics approach, they identified metabolites that can reduce *P. mirabilis* urease activity, a key virulence factor contributing to urinary stone formation and catheter blockage. They identified 7 metabolites that reduced urease activity. D-imidazole lactate, also exhibited antimicrobial properties and reduced urease operon expression. They also proved that the combination of some metabolites with AHA increases the reduction of urease activity, and showed how two of these combinations prevent catheter encrustation in an in vitro CAUTI model. I suggest this paper should be accepted conditional to major revision.

As mentioned in the invitations letter to review, this work is not fully suited to my remit, and my 'expert opinion on the metabolomic analyses in this work would be most valued'.

Therefore, I initially included my comments on the metabolomics analysis, followed by some more general comments:

Metabolomics analysis

The authors use an external service for the untargeted metabolomics analysis (Metabolon Inc's comprehensive Precision Metabolomics TM LC-MS platform for untargeted global metabolomics), however there is very little information provided about the data generated. The mention that 307 metabolites were measured, 262 named metabolites and 45 that could not be assigned a specific chemical structure.

- Where is the data table reporting the measured intensities for each of these compounds? Such data must also contain mass to charge ratios and retention times
- Were MS2 spectra acquired in order to facilitate the annotation.
- The metabolites annotations are provided without any sort of confidence associated with them, nor with any supporting evidence for them. Are these all-putative annotations based on MS1 only, or are they level 1 identifications? Please refer to the guidelines for reporting annotations from the metabolomics standards initiative (MSI).
- The statistical analysis is not described properly: 'p-values for each comparison were derived from the Natural Log-Transformed fold change using one-way ANOVA with Tukey's test for multiple comparison (q=false discovery rate).' The natural log transformed fold changes of what? It is not clear.

Overall, there is very little information that can be used to assess the quality of the metabolomics analysis. They authors should provide all the necessary information, including the data, following the guidelines from the metabolomics society

Additional comments

- The authors assumed that all secreted factors of interests were small molecules, hence the metabolomics approach. Could the authors explain why secreted proteins were not considered?

- Figure 1 is not clear. The choice of grayscale rather than colours make it difficult to read, especially, A, B, and C.
- “Compared to *P. mirabilis* incubated in an autologous supernatant, the urease-modulating phenotypes of *E. faecalis* and *M. morgani* supernatants remained statistically significant following all treatments”
Not clear what is statistically significant from this sentence, and what is the direction of the change observed.
- Can the authors explain why only axenic cultures were considered for the identification of metabolites of interest. It is possible that co-cultures would have elicited the production of different compounds.
- Are the genomes of the bacteria available? Was sequencing considered? Genomes can be mined for secondary metabolites with tools such as antiSMASH

Reviewer #2

(Remarks to the Author)

The manuscript describes identification and characterization of intrinsic metabolites secreted by uropathogenic species that control viability and/or ureolytic activity of *Proteus mirabilis*. The work is a development of previous observations of the research group on the inhibitory effect of *Morganella morgani* live cells on the virulence of *P. mirabilis* and the potential of studying microbial interactions in heterogenic UTIs. The research comprised numerous approaches and analytical methods for elucidation of the activity reduction mechanism including direct inhibitory activity towards urease enzyme, disturbing urease synthesis or toxic effect against *P. mirabilis* cells, the models used included simulated UTI in artificial catheterized bladder system. I find the work important and deserving to be published. Here are my comments and questions to the authors:

1. Would it be possible to estimate the concentration of the selected metabolites in live cultures supernatants that were shown to reduce ureolytic activity in Fig.1?
2. Studies on isolated urease were accessory and served solely to indicate the presence or absence of direct inhibition, still *Canavalia ensiformis* enzyme is not a good reference. Ureases possess highly conserved active site but the affinity towards substrate and overall structure differences influence their susceptibility to inhibitors. *P. mirabilis* urease is not available for purchase but it is produced in abundance in live cells induced with urea and can be easily partially purified.
3. Acetohydroxamate and azoles are competitive urease inhibitors, assays with 500 mM urea present (JBU K_m is ~1.5 mM) most likely result with underestimation of the inhibition level when isolated enzyme is used.
4. AHA and azoles are slow binding inhibitors of urease; instead of following the assay for 90 min. starting from complete reaction mixtures, additional preincubation of enzyme/cells with inhibitor prior to substrate addition would be a better approach. The progress curves are not provided either for JBU or live *Proteus* ureolysis so it is difficult to evaluate if the results reflect maximum inhibition that could be obtained with inhibitors concentration range used.
5. It is not clear to me why D-imidazole lactate has been excluded from in vitro CAUTI model studies and cytotoxicity towards human normal cells estimations. The compounds tested using glass bladder system did not reduce *P. mirabilis* viability so the effects observed were solely due to urease dampening and cooperation with AHA, it is ok, but then they were tested in combination with 3rd generation cephalosporin. In my opinion evaluation of this most interesting metabolite in CAUTI model and assays of its activity possibly additive/synergistic to established antimicrobials would be truly valuable.

Reviewer #3

(Remarks to the Author)

This manuscript by Guterman et al. demonstrates the potential of bacterial metabolites to dampen the urease activity of uropathogens. CAUTI is a significant global public health problem, and this study represents an innovative approach to hamper uropathogen growth on catheters. Building on their observations from polymicrobial UTIs, they utilized untargeted metabolomics to detect urease-dampening metabolites. They describe the effects of these metabolites on uropathogen growth, urease activity, and expression of the urease operon of *P. mirabilis*. Some of these metabolites increase urease inhibition efficacy of AHA. Combinatorial effect on urease dampening and synergy with antimicrobials highlight the potential of these metabolites to guide future drug development efforts. Furthermore, clinical relevance of urease dampening metabolites on catheter encrustation and crystalline biofilm formation was demonstrated using a glass bladder model. Methods described here are sufficient to reproduce the work independently. The Authors have clearly outlined the limitations of the study, including biological activity of urease dampening metabolites at urinary and systemic sites. The manuscript is well-written, logically organized, and conclusions are supported by their results.

Major comments:

- 1) Reason for selecting histamine and 4-imidazole acetate even though there are other combinations like leucylglycine + histamine, phenylpyruvate + imidazole has lesser combination index than histamine + 4 imidazole should be clarified.
- 2) The Authors indicate differences in buffering capacity between AUM and human urine. Demonstrating the urease-dampening and biofilm inhibitory effects of most effective combinations against *P. mirabilis* in human urine will significantly increase the impact of presented findings.
- 3) D-IL has growth inhibitory effect against *P. mirabilis*, and it also decreases abundance of urease operon transcripts. Is there a change in *rpoA* levels during D-IL treatment? Is there a non-specific decrease in transcription or transcript stability during treatment? Kindly discuss.
- 4) Evaluating the biofilm disruption potential of the best combination described in this study using discarded catheters to establish proof-of-principle will elevate the clinical impact of this study.
- 5) D-IL has growth inhibitory effect against *P. mirabilis*. However, only AHA-Histamine was tested for synergy with ceftriaxone. Kindly comment on the rationale for not including D-IL for testing for synergistic antimicrobial activity.

Minor comments:

- 1) Line 107: *P. mirabilis* should be in italics
- 2) 303-307: eyelet, not islet
- 3) 382: site, not sight
- 4) Mention media used in legend for Fig 1D-K

Reviewer #4

(Remarks to the Author)

In Beryl et al. the authors identify metabolites enriched in the supernatants of bacteria that reduce urease activity when co-cultured with *P. mirabilis*. The authors prioritized a set of 34 metabolites that showed >5 fold relative abundance in the supernatants of bacteria that reduced activity relative to the *P. mirabilis* supernatants or those from bacteria that enhanced urease activity. The authors identified 7 metabolites that reduced *P. mirabilis* activity in a dose dependent manner and analyzed the effects of these metabolites on the activity of Jack Bean Urease. They show that among the compounds only D-imidazole lactate reduces viable cell count and affects expression of the urease operon. They further show that the metabolites are effective at reducing urease activity in several other urease producing UTI pathogens and that several pairwise combinations of the metabolites and AHA, an FDA approved urease inhibitor, had synergistic dampening effects. The synergistic effects of histamine and 4-imidazole acetate with AHA were tested in an in vitro bladder catheterization model and were shown to reduce the urease dependent increase in pH and catheter encrustation in the absence of a reduction in *P. mirabilis* CFUs. Finally, the authors show that the combination of AHA and histamine can increase the activity of ceftriaxone on *P. mirabilis* in the in vitro bladder model.

Overall, Beryl et. al is well written and introduces a conceptually interesting and important line of inquiry into antibiotic-sparing therapeutics that can be used for prophylactic management of CAUTI caused by urease-producing pathogens. The manuscript could be improved by addressing the following major considerations:

- 1) The concentrations at which the metabolites are used in these assays are extremely high (mM range). Although the authors show synergistic effects with pairwise combinations of the compounds (Fig 8), these combinations are still used at mM concentrations. This is addressed in the discussion section but remains a critical limitation of the study.
- 2) The authors use Jack Bean Urease in their enzymatic assays throughout the manuscript. Although the active site is 100% conserved with that of *P. mirabilis*, other unique regions of the protein might interact allosterically with the metabolites which would be missed using JBU. Unless purification or monitoring of enzymatic activity in *P. mirabilis* urease is not possible, the authors should purify and use this enzyme in their assays instead.
- 3) The authors investigate the mechanisms of metabolite inhibition and suggest direct vs indirect mechanisms of action. It isn't completely clear what is meant by these terms. For clarity, it would help if the authors defined "direct" as competition with the active site, "indirect" as allosteric inhibition of the enzyme, and urease independent mechanism of dampening for other effects.
- 4) To determine the mechanism of action as defined in (3), the authors should show enzyme kinetic curves, with activity as a function of enzyme concentration in the presence/absence of the inhibitor, to determine Vmax and Km values. The authors can then distinguish between direct (active site) vs indirect (allosteric) inhibition.
- 5) For Supplementary Figure 7, please include an alignment of the *P. mirabilis* HI4320, 104V0, 106V15, and HU1069 urease enzymes.
- 6) In the assessment of the effect of the metabolites on the urease activity in other bacterial strains, the authors should state whether the D-imidazole lactate was killing the Gram-negative bacteria but not *S. aureus*. To this end, please provide CFUs in addition to the growth curves presented in Supplementary Figure 9.
- 7) While the in vitro model system that the authors set up to test catheter encrustation is very interesting, there are several limitations, as noted in the discussion, compared to an in vivo system. One that is particularly important to this study is the effect of the addition of exogenous histamine on the immune system. I am not an immunologist, but my concern is that if histamine is administered prophylactically in vivo it could trigger a more robust immune response that increases the deposition of host factors on the catheter which could increase bacterial colonization. The authors should test this in a murine CAUTI model or explain why these experiments could not be completed.

Minor points:

- 1) The "and/or" statement in "We therefore sought to.....3kDa size exclusion filtrations and/or..." is confusing. Weren't all the treated supernatants in 1B also filtered? Please clarify this sentence.
- 2) Please shorten the section in the main text explaining how the effect of the different imidazole lactate enantiomers was determined. As written, this section is distracting.
- 3) The effect of the metabolites on the pH in the urease assays should be moved to the main text, as this is a major consideration for the mechanism of action.
- 4) How representative of *P. mirabilis* urease enzymes are the selected strains HI4320, 104V0, 106V15, and HU1069?
- 5) Please remove language suggesting trends towards significance throughout the manuscript.
- 6) Please maintain the quotes around "bladder" or refer to the device as the in vitro bladder system throughout the text. There are times when bladder is simply used.

Version 1:

Reviewer comments:

Reviewer #1

(Remarks to the Author)

Overall the authors gave satisfactory answers to most comments, but there are still few questions that are not replied in full.

Reviewer 1 - comment 3: Were MS2 spectra acquired in order to facilitate the annotation? The authors point to the metabolights repository, but do not provide a clear answer. It seems that such data was acquired, but a more direct answer would be appreciated.

Reviewer 1 - comment 4: Using the tier system for expressing the confidence in annotation is fine, but the asterisks system is quite misleading. If an annotation does not meet the requirements for tier 1, it simply has to drop to tier 2. Moreover, there is no clear explanation about the difference between 1 or 2 asterisks. The authors should stick to the tiers defined by the MSI without adding new intermediate tiers.

Reviewer 1 - comment 5: the authors provided a more detailed description of the statistical analysis, which is always good, but they failed to answer the actual question. What are they comparing?

Reviewer #2

(Remarks to the Author)

The revised manuscript by Guterman et al. contains new material that describes the results of additional experiments that were suggested in my review. I am satisfied with the authors' response to minor comments. In my opinion can be published as it is.

Reviewer #3

(Remarks to the Author)

This manuscript by Guterman et al. demonstrates the potential of bacterial metabolites to dampen the urease activity of uropathogens. CAUTI is a significant global public health problem, and this study represents an innovative approach to hamper uropathogen growth on catheters. Building on their observations from polymicrobial UTIs, they utilized untargeted metabolomics to detect urease-dampening metabolites. They describe the effects of these metabolites on uropathogen growth, urease activity, and expression of the urease operon of *P. mirabilis*. Some of these metabolites increase urease inhibition efficacy of AHA. Combinatorial effect on urease dampening and synergy with antimicrobials highlight the potential of these metabolites to guide future drug development efforts. Furthermore, clinical relevance of urease dampening metabolites on catheter encrustation and crystalline biofilm formation was demonstrated using a glass bladder model. Methods described here are sufficient to reproduce the work independently. The Authors have clearly outlined the limitations of the study, including biological activity of urease dampening metabolites at urinary and systemic sites. The manuscript is well-written, logically organized, and conclusions are supported by their results. The Authors have satisfactorily addressed the concerns raised by the Reviewers.

Reviewer #4

(Remarks to the Author)

The manuscript by Beryl et al. entitled "Harnessing microbial-derived metabolites in the urinary tract to prevent infection induced catheter encrustation" is well-written, addresses an issue of significant concern to human health, and advances our understanding of polymicrobial interactions in human infections. The paper addresses catheter-associated urinary tract infections (CAUTI) caused by *Proteus mirabilis* (Pm) and focuses on the bacterium's urease activity. Building on a foundation of previous research showing that polymicrobial interactions can influence Pm urease function, the authors aim to identify novel, small molecule inhibitors to inhibit urease activity in order to address the limited therapeutic options available. The authors perform a broad metabolomic screen to identify metabolites that inhibit Pm urease and then characterize their top candidates using a suite of cellular, microbiological, and biochemical analyses to determine the method of interference. The authors use of an in vitro "bladder" system to measure encrustation is particularly clever. Overall the paper is successful in identifying small molecules that work at the same level as current therapeutic options and can synergize with those therapies as well. The paper's methodology to characterize the effects of their candidate Pm urease inhibitors is broad, meaning that the authors are able to show inhibition across multiple species and urease enzymes. The authors are careful in their interpretation of their data and do not overestimate their findings and acknowledge that their top candidates need further optimization to address issues in translation to human health. The authors have also addressed previous reviewer comments fully and transparently. There are relatively minor weaknesses in this draft of the manuscript, detailed more fully below. One area that could be further addressed is the apparent paradox in the authors' interpretation of the results from the Michaelis-Menten analysis and their findings of significant levels of synergy between urease inhibitors. In lines 187-190, the authors show that their interpretation of the Michaelis-Menten analysis suggests competitive inhibition via direct interactions with the active site of the urease enzyme. However, in Figure 6, the authors find that there is significant synergy between small molecules. I would expect there to be more antagonism or neutral interactions, rather than synergy, between the molecules in this case. The authors may want to address these findings in the Discussion and/or provide additional context to address these seemingly contradictory findings.

Ultimately, this paper sets a meaningful hypothesis and provides a novel finding that advances our understanding of polymicrobial infections. I believe that this manuscript is suitable for publication, with minor edits.

Major Concern:

1. See above for discussion on synergy of small molecule inhibitors despite the apparent competitive inhibition mechanism of action.
2. In Figure 7D, the combination of AHA and 4IA appears to be less effective in modulate pH changes than AHA alone, despite AHA being used at the same concentration in both conditions. Please address this.

Minor Concern:

1. Please change the colors and the symbols in Figure 7 to distinguish between the concentrations of the same molecule. It is difficult to distinguish between the concentrations of the same metabolite. A dotted line and a change of the symbol could be used here.
2. In Figure 7 and 9, do the differences between conditions reach statistical significance at any of the individual time points? In particular, in comparing conditions in Fig 9 (Pm + cef versus Pm + AHA + His + cef), do any of these time points reach significance?
3. Do the authors have an explanation for the loss of antimicrobial activity in human urine for D-imidazole lactate relative to other media?
4. Check for proper capitalization of "Gram", as in Line 529 the word is not capitalized as it should be.
5. I believe that there is a typo and that the authors mean to switch the citation of Supp Figs 21 and 22 (Lines 309 and 314).

I would like to thank the reviewers for their insightful and helpful comments. Please see our responses and the associated added/modified text (highlighted in yellow in the manuscript document).

	Reviewer Comment	Response	Location
1.	Reviewer 1: Overall, there is very little information that can be used to assess the quality of the metabolomics analysis. They authors should provide all the necessary information, including the data, following the guidelines from the metabolomics society	Thank you for your comments regarding the metabolomics dataset and analysis. As requested, in conjunction with Metabolon, we have uploaded all metabolomics data (MZML files stripped of proprietary information) to the MetaboLights database (https://www.ebi.ac.uk/metabolights/reviewer81d00c68-bf4d-4db6-ad69-412dfe00b64d). MetaboLights is a database for Metabolomics experiments and derived information. The database is cross-species, cross-technique and covers metabolite structures and their reference spectra as well as their biological roles, locations and concentrations, and experimental data from metabolic experiments. MetaboLights collaborates closely with major parties in world-wide metabolomics communities, such as the Metabolomics Society and the associated Metabolomics Standards Initiative (MSI). Revised text: Saved supernatants from three independent replicates of each strain were sent to Metabolon Inc for global untargeted metabolomics analysis using the comprehensive Precision Metabolomics™ LI-MS platform (53, 54). Raw data from the main metabolomics experiments are available in the MetaboLights database (https://www.ebi.ac.uk/metabolights/reviewer81d00c68-bf4d-4db6-ad69-412dfe00b64d).	Methods Sample preparation for Global Untargeted Metabolomics Paragraph 2
2.	Reviewer 1:	This data was also uploaded to the MetaboLights database. These data are the calculated area under the curve for the peak associated with each	

	Where is the data table reporting the measured intensities for each of these compounds? Such data must also contain mass to charge ratios and retention times	metabolite. The "unnamed biochemicals" are peaks that Metabolon reliably detected and that have consistent retention times, ion masses, and fragmentation patterns, but they cannot assign a specific biochemical structure to them at this time. These are either truly unknown, or that suitable purified standards are unavailable.	
3.	Reviewer 1: Were MS2 spectra acquired in order to facilitate the annotation	Please see the uploaded MetaboLights database (https://www.ebi.ac.uk/metabolights/reviewer81d00c68-bf4d-4db6-ad69-412dfe00b64d).	
4.	Reviewer 1: The metabolites annotations are provided without any sort of confidence associated with them, nor with any supporting evidence for them. Are these all-putative annotations based on MS1 only, or are they level 1 identifications? Please refer to the guidelines for reporting annotations from the metabolomics standards initiative (MSI).	All reported biochemicals in Sup. Table 7 that are given without an asterix (*) are considered Tier 1 identifications according to the Metabolomics Standards Initiative. Metabolites reported with either 1 (*) or 2 (**) asterisks do not officially meet the Tier 1 requirements, but do not drop to the Tier 2 level. Metabolon considers them 'Not Tier 1' identifications. Most often this is due to a lack of purified standard that they can compare to directly, while they are confident in the identity based on other information. Added text: All reported biochemicals in Sup. Table 7 that are given without an asterix (*) are considered Tier 1 identifications according to the Metabolomics Standards Initiative. Metabolites reported with either 1 (*) or 2 (**) asterisks do not officially meet the Tier 1 requirements, but do not drop to the Tier 2 level. Therefore, they are designated as 'Not Tier 1' identifications.	Methods Sample preparation for Global Untargeted Metabolomics Paragraph 2

5.	Reviewer 1: The statistical analysis is not described properly: ‘p-values for each comparison were derived from the Natural Log-Transformed fold change using one-way ANOVA with Tukey’s test for multiple comparison (q=false discovery rate).’ The natural log transformed fold changes of what? It is not clear.	We have clarified the language describing the Metabolon platform and statistical analysis for the metabolomics data. Please see the revised text below and highlighted in the revised manuscript. Revised Text: The measured intensities for each metabolite is the calculated area under the curve for the peak associated with each metabolite. “Peak Area Data” is scaled to the median value for each metabolite (all samples are ranked low to high; the middle value is set to 1.0 and all other values are scaled accordingly). This does not change the distribution of the data other than setting all metabolites on a similar scale. The median-scaled data are referred to as “Batch norm data.” Any missing values in the median-scaled data are then imputed with the minimum value detected across the dataset (54). As the Metabolon platform is optimized for the broadest metabolite coverage, the limit of detection (LOD) for any given compound is not the same and not necessarily optimized. Therefore, the percentage of samples in each bacterial cell free supernatant sample that have a metabolite detected is given as the "% Fill" in Sup Table 7. The “Batch norm data” are then natural log transformed for the statistical analysis. “Fold change” was calculated as the ratio of the mean scaled intensity for a given metabolite between two experimental groups (Sup. Table 7). Fold change values for each group shown in Sup. Table 7 uses the batch-normalized imputed values. All reported biochemicals in Sup. Table 7 that are given without an asterix (*) are considered Tier 1 identifications according to the Metabolomics Standards Initiative. Metabolites reported with either 1 (*) or 2 (**) asterisks do not officially meet the Tier 1 requirements, but do not drop to the Tier 2 level. Therefore, they are designated as 'Not Tier 1' identifications. p-values for each comparison were derived from Batch-Norm Imputed values using one-way ANOVA with Tukey’s test for multiple comparison (q=false discovery rate).	Methods Sample preparation for Global Untargeted Metabolomics Paragraph 2
-----------	--	--	--

6.	Reviewer 1: The authors assumed that all secreted factors of interests were small molecules, hence the metabolomics approach. Could the authors explain why secreted proteins were not considered?	We filtered our cell free supernatants using a 3kD filter and repeated our urease activity assays using whole cell P. mirabilis. Since the average amino acid is 110 Da, a 3kD filter would filter out the secreted proteins that are approximately > 30 amino acids. Since the 3kD filtered cell free supernatants of M. morganii and E. faecalis maintained their urease dampening and enhancing phenotypes respectively, it suggested that the secreted urease modulating factors were small molecules or small secreted proteins that are less than 30 amino acids. Given the narrow size range for proteomics and that the secreted factors were heat stable, we decided to pursue an untargeted metabolomics approach to screen for urease modulating metabolites. Added text: Since the 3kD filtered cell free supernatants of M. morganii and E. faecalis maintained their urease dampening and enhancing phenotypes respectively, the secreted urease modulating factors are also likely to be small molecules or small secreted proteins that are less than 30 amino acids. Given the narrow size range for proteomics and that the secreted factors were heat stable (Fig 1B), we decided to pursue an untargeted metabolomics approach to screen for urease modulating metabolites.	RESULTS Co-colonizing species modulate P. mirabilis urease activity via secreted small molecules. Paragraph 1
7.	Reviewer 1: Figure 1 is not clear. The choice of grayscale rather than colures make it difficult to read, especially, A, B, and C.	We have revised all figures, so they are colored and no longer grayscale.	
8.	Reviewer 1: “Compared to P. mirabilis incubated in an autologous supernatant,	Please see the revised text. Revised Text: Compared to P. mirabilis incubated in an autologous supernatant, P. mirabilis incubated in E. faecalis supernatant exhibited a statistically	RESULTS Co-colonizing species modulate P.

	the urease-modulating phenotypes of E. faecalis and M. morganii supernatants remained statistically significant following all treatments” Not clear what is statistically significant from this sentence, and what is the direction of the change observed.	significant enhancement of urease activity (higher AUC), whereas P. mirabilis incubated in M. morganii supernatant exhibited a statistically significant reduction in urease activity (lower AUC). These urease-modulating effects remained statistically significant following all treatments (Fig1. B), indicating that the urease-modulatory factors are likely to be heat-stable molecules that do not require metal cofactors or act as metallophores for the supply or sequestration of nickel from the urease apoenzyme.	mirabilis urease activity via secreted small molecules. Paragraph 1
9.	Reviewer 1: Can the authors explain why only axenic cultures were considered for the identification of metabolites of interest. It is possible that co-cultures would have elicited the production of different compounds.	We previously demonstrated that several uropathogens, including E. faecalis and P. stuartii, can enhance P. mirabilis urease activity during coculture in urine while M. morganii appears to dampen activity (1, 2). We also demonstrated that P. mirabilis urease activity was significantly increased during incubation in cell-free spent urine supernatants from E. faecalis and P. stuartii, compared to the level of activity for incubation in supernatant from the isogenic Δure mutant, and the activity was significantly reduced during incubation in supernatant from M. morganii. These findings indicate that this urease modulation by these species is mediated by a soluble factor present in spent media and is independent of direct cell-cell contact or proximity of bacteria (1). However, it is possible for co-cultures to produce additional metabolites that can modulate P. mirabilis urease activity.	
10.	Reviewer 1: Are the genomes of the bacteria available? Was sequencing considered? Genomes can be mined for secondary metabolites with tools such as antiSMASH	At least one bacterial isolate for each species’ cell free supernatant that was sent for metabolomics has a complete genome sequence available. For future work on this project, we plan to take this approach of mining for secondary metabolites using these genomes.	

11.	Reviewer 2: Would it be possible to estimate the concentration of the selected metabolites in live cultures supernatants that were shown to reduce ureolytic activity in Fig.1?	Unfortunately, it is not possible to estimate the concentration of the selected metabolites in live cultures supernatants using the metabolomics data. The global untargeted platform used in this study is designed to capture as many metabolites as possible, so it is not optimized for quantitative detection of a single metabolite or even a class of metabolites. The various extraction and ionization efficiency for any given biochemical varies widely in a given matrix. When Metabolon adds compounds to their library, they are run at multiple concentrations, but even these values are optimal since they use purified compounds rather than a compound in a matrix that also contains thousands of others. Within a given experiment, Metabolon can utilize relative levels to compare groups, but relating these back to the concentrations in the original sample is not feasible as it would require standard curves for every compound that is detected, run in addition to the samples of interest. Something we aim to do in the future is to examine the concentration of urease dampening metabolites (both over time and under different culture conditions) to maximize their effective dose to modulate bacterial urease activity. However, this is beyond the scope of this work.	
12.	Reviewer 2: Studies on isolated urease were accessory and served solely to indicate the presence or absence of direct inhibition, still Canavalia ensiformis enzyme is not a good reference. Ureases possess highly	Thank you for your suggestion. We have repeated our urease activity assays using the cell free extracts of P. mirabilis HI4320. We have replaced the JBU results in Figure 2 with the urease activity of cell free extracts of P. mirabilis data and added the following text below. Added/revised text: To assess whether dampening metabolites directly interact with the urease enzyme, we repeated the urease activity assay with cell free P. mirabilis extracts instead of whole-cell P. mirabilis. The FDA-approved urease inhibitor acetohydroxamic acid (AHA) was used as a positive control for all experiments, as AHA is a urea analogue that directly interacts with the urease active site. Histamine, leucylglycine, D-imidazole lactate, L-	RESULTS Histamine, Leucylglycine, D-imidazole lactate, L-imidazole lactate, imidazole and 4-imidazole acetate dampen

	conserved active site but the affinity towards substrate and overall structure differences influence their susceptibility to inhibitors. P. mirabilis urease is not available for purchase but it is produced in abundance in live cells induced with urea and can be easily partially purified.	imidazole lactate, imidazole and 4-imidazole acetate were all capable of at least partially inhibiting the urease activity of cell free P. mirabilis extracts while phenylpyruvate did not, suggesting a mechanism of action independent of the catalytically active urease enzyme for phenylpyruvate (Fig. 2). Analysis of Michaelis–Menten parameters in the presence and absence of urease dampening metabolites was conducted to indicate the type of inhibition (Sup. Fig. 7-12). The estimated Vmax remained approximately the same upon addition of histamine, leucylglycine, L-imidazole lactate, imidazole and 4-imidazole acetate, while the estimated Km increased suggesting these metabolites act as competitive inhibitors by competing directly with the substrate for the active site (Sup. Fig. 7-12). In contrast, the estimated Vmax was reduced while the Km remained unaffected upon the addition of D-imidazole lactate, suggesting a noncompetitive or mixed method of inhibition (Sup. Fig 9). Analysis of Michaelis–Menten parameters was not possible for phenylpyruvate as this metabolite did not inhibit urease activity in cell free P. mirabilis extracts.	urease activity in cell free P. mirabilis extracts. Paragraph 2
13.	Reviewer 2: Acetohydroxamate and azoles are competitive urease inhibitors, assays with 500 mM urea present (JBU Km is ~1.5 mM) most likely result with underestimation of the inhibition level when isolated enzyme is used.	We chose to use 500 mM urea in these assays because this concentration allows us to detect differences between conditions in the in vitro 'glass bladder' model. We have repeated these urease activity assays with a wider range of urea concentrations for the analysis of Michaelis–Menten parameters in the presence and absence of each urease dampening metabolite (See Sup. Fig 7-12). These experiments were performed using cell free P. mirabilis extracts to indicate the type of enzyme inhibition. Please see the response to reviewer comment #12 for details regarding these experiments.	
14.	Reviewer 2: AHA and azoles are slow binding inhibitors	We have added the progress curves for the urease activity assays with whole cell P. mirabilis (Sup. Fig. 2), cell free extracts of P. mirabilis (Sup. Fig. 6) and purified JBU (Sup. Fig. 14). While we agree that incubation with the inhibitors prior to addition of substrate would be a more robust approach for	

	of urease; instead of following the assay for 90 min. starting from complete reaction mixtures, additional preincubation of enzyme/cells with inhibitor prior to substrate addition would be a better approach. The progress curves are not provided either for JBU or live Proteus ureolysis so it is difficult to evaluate if the results reflect maximum inhibition that could be obtained with inhibitors concentration range used	further examining effective inhibitor concentrations, this would not be possible in the translational models as substrate is present for the duration of the experiment (as it would be in the urinary tract). This is something we aim to explore in our future studies towards designing improved inhibitors.	
15.	Reviewer 2: It is not clear to me why D-imidazole lactate has been excluded from in vitro CAUTI model studies and cytotoxicity towards human normal cells estimations. The compounds tested using glass bladder system did not reduce P. mirabilis viability so the effects	We appreciate the reviewer’s thoughtful comment and agree that D-imidazole lactate is a promising candidate for further study, particularly regarding its potential synergistic effects with antimicrobials in the CAUTI model. At this stage, we prioritized metabolites that demonstrated cooperation with AHA as this urease inhibitor has proven to be clinically effective in reducing catheter encrustation. We also encountered issues with obtaining sufficient quantities of pure D-imidazole lactate, and therefore had to excluded testing of this metabolite in the glass “bladder” system due to limited availability. However, we did perform the cytotoxicity assays with D-imidazole lactate and the remaining metabolites and found no decrease in percent viability is both cell lines (see revised text below). Moreover, future work will focus on structure-activity relationship (SAR) studies to improve metabolites efficacy in dampening urease activity.	RESULTS Histamine and 4-imidazole acetate synergize with AHA to reduce catheter encrustation in an in vitro

	observed were solely due to urease dampening and cooperation with AHA, it is ok, but then they were tested in combination with 3rd generation cephalosporin. In my opinion evaluation of this most interesting metabolite in CAUTI model and assays of its activity possibly additive/synergistic to established antimicrobials would be truly valuable.	Added/revised text: Urease dampening metabolites were assessed for potential cytotoxic effects in monolayers of a human embryonic kidney cell line (HEK293) and a bladder epithelial cell line (T24) using an enzymatic cytotoxicity assay. None of the conditions tested demonstrated any reduction in viability of either cell line after a 24-hour incubation period (Sup. Fig. 22).	CAUTI model. Paragraph 1
16.	Reviewer 3: Reason for selecting histamine and 4-imidazole acetate even though there are other combinations like leucylglycine + histamine, phenylpyruvate + imidazole has lesser combination index than histamine + 4 imidazole should be clarified.	The criteria for selecting urease dampening metabolites for further testing with AHA in the clinically representative in vitro “bladder” model of catheter encrustation includes: a significant combination index, a low effective dose to achieve a significant combination index and prioritizing combinations that reduce the effective dose of AHA when tested in AUM. Leucylglycine and phenylpyruvate required concentrations >50mM to achieve a significant combination index with all metabolites. Therefore, they were not chosen to test further in the in vitro “bladder” model. Revised text: Histamine and 4-imidazole acetate were selected for further testing with AHA in a clinically-representative in vitro “bladder” model of catheter encrustation (28) as these two urease-dampening metabolites exhibited significant synergistic inhibition of P. mirabilis urease activity (CI<1, Fig 6)	RESULTS Histamine and 4-imidazole acetate synergize with AHA to reduce catheter encrustation in an in vitro CAUTI model.

		at concentrations <50mM (Sup. Table 5) and they exhibited the greatest reduction in the required dose of AHA when tested in AUM (Sup. Fig. 23).	Paragraph 3
17.	Reviewer 3: The Authors indicate differences in buffering capacity between AUM and human urine. Demonstrating the urease-dampening and biofilm inhibitory effects of most effective combinations against P. mirabilis in human urine will significantly increase the impact of presented findings.	We have repeated our whole cell P. mirabilis urease activity assays in undiluted human urine supplemented with 500mM urea. Although there is a lower buffering capacity in human urine compared to the potassium phosphate buffer and artificial urine media all metabolites demonstrated less potent activity in human urine. This finding would suggest some compound(s) within human urine is interfering with the activity of these metabolites against P. mirabilis urease. We have added this data as Supplemental Figure 21. Please see the added text to the manuscript below. Added text: All seven urease-dampening metabolites reproducibly decreased P. mirabilis urease activity in a dose dependent manner, although less-pronounced, in human urine (Sup. Fig 21).	RESULTS Histamine and 4-imidazole acetate synergize with AHA to reduce catheter encrustation in an in vitro CAUTI model. Paragraph 1
18.	Reviewer 3: D-IL has growth inhibitory effect against P. mirabilis, and it also decreases abundance of urease operon transcripts. Is there a change in rpoA levels during D-IL treatment? Is there a non-specific decrease in transcription or transcript stability during treatment? Kindly discuss.	Incubating P. mirabilis in D-imidazole lactate for 15 minutes did not affect the pool of rpoA mRNA transcripts. We have included the raw Cq values for all conditions and primers, including rpoA, in Sup. Fig. 16. We also performed a RNA integrity analysis using agarose gel electrophoresis on our treated and nontreated extracted RNA. For all samples, the gels demonstrated three distinct bands with no smearing indicating no change in transcript stability. Added text: For analysis, rpoA was chosen as the reference gene for normalization between samples as it exhibited low variation in the presence and absence of urea (Sup Fig 16B-H). A RNA integrity analysis using agarose gel electrophoresis on extracted RNA was performed for all samples to confirm no change in transcript stability (Sup Fig 16A). The data were analyzed	MATERIALS AND METHODS Real-time quantitative PCR Paragraph 1

		according to the Relative Quantification (RQ) method by Pfaffl et al 2001 (56) (See Supplemental Materials).	
19.	Reviewer 3: Evaluating the biofilm disruption potential of the best combination described in this study using discarded catheters to establish proof-of-principle will elevate the clinical impact of this study.	We agree it would be interesting to look at this as part of our future work. However, we look at these urease dampening metabolites as a prophylactic approach and therefore it is unlikely to reduce an established biofilm.	
20.	Reviewer 3: D-IL has growth inhibitory effect against P. mirabilis. However, only AHA-Histamine was tested for synergy with ceftriaxone. Kindly comment on the rationale for not including D-IL for testing for synergistic antimicrobial activity.	Please see our response to comments #15 and #16	
21.	Reviewer 3 (minor comment): Line 107: P. mirabilis should be in italics	We have corrected this.	
22.	Reviewer 3 (minor comment):	We have corrected this.	

	303-307: eyelet, not islet		
23.	Reviewer 3 (minor comment): 382: site, not sight	We have corrected this.	
24.	Reviewer 3 (minor comment): Mention media used in legend for Fig 1D-K	Revised text: (D-K) Urease activity of Pm incubated in candidate urease-dampening metabolites or the acetohydroxamic acid (AHA) positive control in potassium phosphate buffer	RESULTS Figure 1 legend
25.	Reviewer 4: The concentrations at which the metabolites are used in these assays are extremely high (mM range). Although the authors show synergistic effects with pairwise combinations of the compounds (Fig 8), these combinations are still used at mM concentrations. This is addressed in the discussion section but remains a critical limitation of the study.	We agree with the reviewer that the concentrations of all metabolites are still too high for translation to clinical use. However, the objective of this study was to provide proof of principle that microbial metabolites can be harnessed towards a more effective therapy to prevent catheter encrustation and blockage. The identified metabolites will therefore provide a strong foundation for SAR studies and a medicinal chemistry approach towards improving efficacy, as mentioned in the discussion.	
26.	Reviewer 4:	Please see our response to comment #12	

	The authors use Jack Bean Urease in their enzymatic assays throughout the manuscript. Although the active site is 100% conserved with that of P. mirabilis, other unique regions of the protein might interact allosterically with the metabolites which would be missed using JBU. Unless purification or monitoring of enzymatic activity in P. mirabilis urease is not possible, the authors should purify and use this enzyme in their assays instead.		
27.	Reviewer 4: The authors investigate the mechanisms of metabolite inhibition and suggest direct vs indirection mechanisms of action. It isn't completely clear what is meant by these terms. For clarity, it would help if the authors	We have clarified the language such that the mechanism of action is defined as competitive, noncompetitive (or mixed), or urease independent mechanism of dampening. See our response to reviewer comment #12 for the revised and added text in the manuscript.	

	defined “direct” as competition with the active site, “indirect” as allosteric inhibition of the enzyme, and urease independent mechanism of dampening for other effects.		
28.	Reviewer 4: To determine the mechanism of action as defined in (3), the authors should show enzyme kinetic curves, with activity as a function of enzyme concentration in the presence/absence of the inhibitor, to determine Vmax and Km values. The authors can then distinguish between direct (active site) vs indirect (allosteric) inhibition.	We have repeated these urease activity assays for the analysis of Michaelis–Menten parameters in the presence and absence of each urease dampening metabolite. These experiments were performed using cell free P. mirabilis extracts to indicate the type of enzyme inhibition. Please see the response to reviewer comment #12 for details regarding these experiments.	
29.	Reviewer 4: For Supplementary Figure 7, please include an alignment of the P. mirabilis HI4320,	We have included the sequence alignment for Pm HI4320, Pm104V0 and Pm106V15 as these are the isolates we have sequencing data on. For all urease operon subunits there is greater than 99% amino acid identity across all three strains. We have added the alignment for all urease operon subunits as Sup. Fig 18 . Please see the added manuscript text below.	RESULTS Dampening metabolites have activity against other

	104V0, 106V15, and HU1069 urease enzymes.	Added/revised text: To determine if the activity of urease-dampening metabolites was strain-specific against P. mirabilis HI4320, we tested dampening capacity against three other P. mirabilis clinical urinary isolates; one from an uncatheterized patient (HU1069) (26) and two from different participants in a recent prospective cohort study of long-term catheterized nursing home residents (104V0 and 106V15, whose urease operon have >99% amino acid identity with P. mirabilis HI4320, Sup. Fig. 18) (7).	urease positive bacteria. Paragraph 1
30.	Reviewer 4: In the assessment of the effect of the metabolites on the urease activity in other bacterial strains, the authors should state whether the D-imidazole lactate was killing the Gram-negative bacteria but not S. aureus. To this end, please provide CFUs in addition to the growth curves presented in Supplementary Figure 9.	We have provided the CFU data in Sup. Fig. 20 for both gram negative (M. morganii TA43 and P. stuartii BE2467) and gram-positive bacteria (UTI MRSA) incubated in serial dilutions of D-imidazole lactate. Based on these data, D-imidazole lactate exhibited robust antimicrobial activity against M. morganii TA43 and P. stuartii BE2467 during incubation in potassium phosphate and less-pronounced antimicrobial activity against UTI MRSA. Please see the added text interpreting these data in the manuscript below. Revised text: The effect of each metabolite on the growth and viability of these urease positive uropathogen species was also assessed. D-imidazole lactate was the only metabolite that perturbed growth, and it also demonstrated antimicrobial activity against each of the tested species (Sup. Fig. 20). Interestingly, the antimicrobial activity of D-imidazole against UTI MRSA was modest, with only a one log reduction in CFUs after one hour incubation in potassium phosphate buffer (Sup Fig 20).	RESULTS Dampening metabolites have activity against other urease positive bacteria. Paragraph 2
31.	Reviewer 4: While the in vitro model system that the authors set up to test catheter encrustation is very interesting, there are several limitations, as noted in the discussion,	We appreciate the reviewer's thoughtful comment and agree that exogenous histamine is a potent inflammatory mediator and has been implicated in the pathogenesis of various lower urinary tract disorders (as mentioned in our discussion). Since the concentrations at which the metabolites are used in these assays are in the millimolar range we believe that future work should first focus on structure-activity relationship (SAR) studies to improve metabolites efficacy and reduce their effective dose in dampening urease activity before exploring their effects in a murine CAUTI model.	

	compared to an in vivo system. One that is particularly important to this study is the effect of the addition of exogenous histamine on the immune system. I am not an immunologist, but my concern is that if histamine is administered prophylactically in vivo it could trigger a more robust immune response that increases the deposition of host factors on the catheter which could increase bacterial colonization. The authors should test this in a murine CAUTI model or explain why these experiments could not be completed.		
32.	Reviewer 4 (minor comment): The “and/or” statement in “We therefore sought to.....3kDa size exclusion filtrations and/or...” is confusing.	We have clarified this language and removed the term “or” as all treatment groups were filtered with 3kDa size exclusion filtration. Revised text: We therefore sought to further characterize the urease-modulating factors by subjecting cell-free bacterial supernatants to a series of treatments prior to inoculation with live, whole-cell P. mirabilis: 3 kDa size-exclusion filtration	RESULTS Co-colonizing species modulate P. mirabilis urease activity via

	Weren't all the treated supernatants in 1B also filtered? Please clarify this sentence	and 1) boiling for 10 min, 2) five freeze-thaw cycles, 3) metal chelation (Chelex), and 4) supplementation with excess nickel (Fig 1. A, B).	secreted small molecules. Paragraph 1
33.	Reviewer 4 (minor comment): Please shorten the section in the main text explaining how the effect of the different imidazole lactate enantiomers was determined. As written, this section is distracting.	We have shortened this section in the main text as requested and moved some of analysis detail into the Supplemental method section. Please see the revised text below. Revised text: To confirm chemical identities, both lot numbers were analyzed by Liquid chromatography high-resolution mass spectrometry (LC-HRMS), proton nuclear magnetic resonance (¹ H-NMR) spectroscopy, and polarimetry (see details in Sup Methods). These analyses support that both lots are imidazole lactate (Sup. Fig. 3C-D, 4A-B) and the two imidazole lactate forms are enantiomers likely resulting in differing urease modulation activity; where D-imidazole lactate had a more potent activity profile compared to the L-imidazole lactate (Sup. Fig. 3).	RESULTS Seven microbial-derived metabolites reproducibly dampen P. mirabilis urease activity. Paragraph 1
34.	Reviewer 4 (minor comment): The effect of the metabolites on the pH in the urease assays should be moved to the main text, as this is a major consideration for the mechanism of action.	We have moved this figure to the main text as Figure 3 .	Figure 3
35.	Reviewer 4 (minor comment):	Please see our response to comment #29	

	How representative of P. mirabilis urease enzymes are the selected strains HI4320, 104V0, 106V15, and HU1069?		
36.	Reviewer 4 (minor comment): Please remove language suggesting trends towards significance throughout the manuscript.	We have removed all language referencing trends towards statistical significance.	
37.	Reviewer 4 (minor comment): Please maintain the quotes around “bladder” or refer to the device as the in vitro bladder system throughout the text. There are times when bladder is simply used.	We have made these changes throughout the manuscript.	

1. B. S. Learman, A. L. Brauer, K. A. Eaton, C. E. Armbruster, A rare opportunist, *Morganella morganii*, decreases severity of polymicrobial catheter-associated urinary tract infection. *Infection and immunity* **88**, e00691-00619 (2019).
2. C. Armbruster *et al.*, Urease Activity is Enhanced During Coculture of Common Catheter-Associated Urinary Tract Infection (CAUTI) Pathogens and Contributes to Severity of Disease in a Murine Infection Model. *Open Forum Infectious Diseases* **3**, (2016).

REVIEWER COMMENTS

Reviewer #1 (Remarks to the Author):

Overall the authors gave satisfactory answers to most comments, but there are still few questions that are not replied in full.

We apologize for any lack of clarity, and have provided all the requested information as detailed below.

Reviewer 1 - comment 3: Were MS2 spectra acquired in order to facilitate the annotation? The authors point to the metabolights repository, but do not provide a clear answer. It seems that such data was acquired, but a more direct answer would be appreciated.

We apologize for the oversight in our prior response to the reviewer. Yes, MS2 spectra were acquired, and this line of the methods has been revised to state “MS2 spectra were acquired, and raw data from the main metabolomics experiments can be accessed in the MetaboLights database.” The Metabolon software uses retention time, mass of the ion, and ion fragmentation pattern to match a peak within the client samples with our library of >5000 metabolites. All identified peaks are matched to the library by the Metabolon software, and also verified by their team of human curators who compare each call to Process Blanks (water) to ensure the peaks are not present as background and to the combined CMTRX (client matrix pool) of samples (positive controls). Metabolon also run at least 1 sample of an internal EDTA plasma sample that they have extensively characterized to help align retention times to account for slight machine variability in each run.

Reviewer 1 - comment 4: Using the tier system for expressing the confidence in annotation is fine, but the asterisks system is quite misleading. If an annotation does not meet the requirements for tier 1, it simply has to drop to tier 2. Moreover, there is no clear explanation about the difference between 1 or 2 asterisks. The authors should stick to the tiers defined by the MSI without adding new intermediate tiers.

The tier designations and asterisks were generated by Metabolon Inc rather than the authors. We therefore did not want to alter the designations. To respond to this comment in the prior review, we reached out to Scott McCulloch, Principal Study Director at Metabolon, and he provided the explanation that was included in our prior response. It is strictly a way for them to indicate that a biochemical did not officially meet that Tier 1 requirements due to lack of a purified standard for direct comparison, but that confidence in the identity of the biochemical is very high based on other information. We have revised this section to state that metabolites with asterisks “do not officially meet the Tier 1 requirements, but do not drop to the Tier 2 level as there is sufficient other information to provide confidence in the identity of the biochemical.”

Reviewer 1 - comment5: the authors provided a more detailed description of the statistical analysis, which is always good, but they failed to answer the actual question. What are they comparing?

The statistical analyses essentially compare the peak area data between a test supernatant of interest and the control *P. mirabilis* supernatants. However, the peak area data were first normalized and then log transformed. We have revised the text of the methods for clarity to state that “Batch norm data were then natural log transformed for calculation of fold change between groups as well as

statistical analysis to identify biochemicals that were significantly over- or under-represented in a given group compared to the control samples by one-way ANOVA with Tukey's test for multiple comparison (q =false discovery rate)."

Reviewer #2 (Remarks to the Author):

The revised manuscript by Guterman et al. contains new material that describes the results of additional experiments that were suggested in my review. I am satisfied with the authors' response to minor comments. In my opinion can be published as it is.

We thank the reviewer for their comments and assessment.

Reviewer #3 (Remarks to the Author):

This manuscript by Guterman et al. demonstrates the potential of bacterial metabolites to dampen the urease activity of uropathogens. CAUTI is a significant global public health problem, and this study represents an innovative approach to hamper uropathogen growth on catheters. Building on their observations from polymicrobial UTIs, they utilized untargeted metabolomics to detect urease-dampening metabolites. They describe the effects of these metabolites on uropathogen growth, urease activity, and expression of the urease operon of *P. mirabilis*. Some of these metabolites increase urease inhibition efficacy of AHA. Combinatorial effect on urease dampening and synergy with antimicrobials highlight the potential of these metabolites to guide future drug development efforts. Furthermore, clinical relevance of urease dampening metabolites on catheter encrustation and crystalline biofilm formation was demonstrated using a glass bladder model. Methods described here are sufficient to reproduce the work independently. The Authors have clearly outlined the limitations of the study, including biological activity of urease dampening metabolites at urinary and systemic sites. The manuscript is well-written, logically organized, and conclusions are supported by their results. The Authors have satisfactorily addressed the concerns raised by the Reviewers.

We thank the reviewer for their comments and assessment.

Reviewer #4 (Remarks to the Author):

The manuscript by Beryl et al. entitled "Harnessing microbial-derived metabolites in the urinary tract to prevent infection induced catheter encrustation" is well-written, addresses an issue of significant concern to human health, and advances our understanding of polymicrobial interactions in human infections. The paper addresses catheter-associated urinary tract infections (CAUTI) caused by *Proteus mirabilis* (Pm) and focuses on the bacterium's urease activity. Building on a foundation of previous research showing that polymicrobial interactions can influence Pm urease function, the authors aim to identify novel, small molecule inhibitors to inhibit urease activity in order to address the limited therapeutic options available. The authors perform a broad metabolomic screen to identify metabolites that inhibit Pm urease and then characterize their top candidates using a suite of cellular, microbiological, and biochemical analyses to determine the method of interference. The authors use of an in vitro "bladder" system to measure encrustation is particularly clever.

Overall the paper is successful in identifying small molecules that work at the same level as current therapeutic options and can synergize with those therapies as well. The papers methodology to characterize the effects of their candidate Pm urease inhibitors is borad, meaning that the authors are

able to show inhibition across multiple species and urease enzymes. The authors are careful in their interpretation of their data and do not overestimate their findings and acknowledge that their top candidates need further optimization to address issues in translation to human health. The authors have also addressed previous reviewer comments fully and transparently.

We thank the reviewer for this assessment.

There are relatively minor weaknesses in this draft of the manuscript, detailed more fully below. One area that could be further addressed is the apparent paradox in the authors' interpretation of the results from the Michaelis-Menten analysis and their findings of significant levels of synergy between urease inhibitors. In lines 187-190, the authors show that their interpretation of the Michaelis-Menten analysis suggests competitive inhibition via direct interactions with the active site of the urease enzyme. However, in Figure 6, the authors find that there is significant synergy between small molecules. I would expect there to be more antagonism or neutral interactions, rather than synergy, between the molecules in this case. The authors may want to address these findings in the Discussion and/or provide additional context to address these seemingly contradictory findings. Ultimately, this paper sets a meaningful hypothesis and provides a novel finding that advances our understanding of polymicrobial infections. I believe that this manuscript is suitable for publication, with minor edits.

We thank the reviewer for this assessment, and have addressed the key comments below.

Major Concern:

1. See above for discussion on synergy of small molecule inhibitors despite the apparent competitive inhibition mechanism of action.

We greatly appreciate the reviewer's comment. Upon re-examining the Michaelis-Menten analyses, we realized that the metabolites previously thought to be competitive inhibitors were actually causing a slight though consistent reduction in V_{max} coupled with a slight though consistent increase in K_m , which is consistent with mixed inhibition rather than competitive inhibition. We have therefore revised the text as follows: "The addition of either histamine, leucylglycine, L-imidazole lactate, imidazole, or 4-imidazole acetate caused a decrease in the estimated V_{max} , an increase in the estimated K_m , and lack of intersection on either the x or y axis of the Lineweaver-Burke plot, suggesting that each of these metabolites act as mixed inhibitors (Sup. Fig. 7-12). In contrast, the addition of D-imidazole lactate caused the estimated V_{max} to decrease, the K_m to remain roughly unchanged, and intersection on the x-axis of the Lineweaver-Burke plot, suggesting non-competitive inhibition (Sup. Fig 9)."

We have also modified the text throughout to reflect this change.

2. In Figure 7D, the combination of AHA and 4IA appears to be less effective in modulate pH changes than AHA alone, despite AHA being used at the same concentration in both conditions. Please address this.

We have revised the colors and used dashed connection lines to emphasize the concentrations of each metabolite alone that were used in the combination treatment. The

combination of 1.25mM AHA and 12.5mM 4IA significantly delayed the rise to a pH greater than 7 (most evident at the 3 hour timepoint), while treatment with either 1.25mM AHA alone (now with dashed lines) or 12.5mM 4IA did not. Treatment with 5mM AHA was, in general, more effective than the combination treatment, but treatment with 1.25mM AHA was not.

Minor Concern:

1. Please change the colors and the symbols in Figure 7 to distinguish between the concentrations of the same molecule. It is difficult to distinguish between the concentrations of the same metabolite. A dotted line and a change of the symbol could be used here.

We appreciate the suggestion and have revised the figure to more clearly distinguish the concentrations by changing the colors and adding dashed lines.

2. In Figure 7 and 9, do the differences between conditions reach statistical significance at any of the individual time points? In particular, in comparing conditions in Fig 9 (Pm + cef versus Pm + AHA + His + cef), do any of these time points reach significance?

We thank the reviewer for this question, and have added asterisks to the bottom of panels 7C and 7D to demonstrate statistical significance of the combination treatment versus the single treatments. We have similarly added asterisks to panel 8B to demonstrate significance of synergy+ceftriaxone versus ceftriaxone alone. This information has also been added to the figure legends.

3. Do the authors have an explanation for the loss of antimicrobial activity in human urine for D-imidazole lactate relative to other media?

The reviewer brings up an excellent point, and we do not yet know the cause of this. One hypothesis is that D-imidazole lactate may work best on bacteria that are growing and dividing more quickly, and growth rate of *P. mirabilis* is slightly slower in human urine than in LB. This is an area of interest for further exploration of D-imidazole lactate.

4. Check for proper capitalization of "Gram", as in Line 529 the word is not capitalized as it should be.

We have ensured all instances of "Gram" are now capitalized.

5. I believe that there is a typo and that the authors mean to switch the citation of Supp Figs 21 and 22 (Lines 309 and 314).

We thank the reviewer for catching this and have swapped the order of Supplemental Figures 21 and 22 to match the text.